# TANK prevents IFN-dependent fatal diffuse alveolar hemorrhage by suppressing DNA-cGAS aggregation

Atsuko Wakabayashi, Masanori Yoshinaga ⓘ, Osamu Takeuchi ⓘ

**Diffuse alveolar hemorrhage (DAH) is one of the serious complications associated with systemic lupus erythematosus, an autoimmune disease whose pathogenesis involves type I IFNs and cytokines. Here, we show that TANK, a negative regulator of the NF-κB signaling via suppression of TRAF6 ubiquitination, is critical for the amelioration of fatal DAH caused by lung vascular endothelial cell death in a mouse model of systemic lupus erythematosus. The development of fatal DAH in the absence of TANK is mediated by type I IFN signaling, but not IL-6. We further uncover that STING, an adaptor essential for the signaling of cytoplasmic DNA sensor cyclic GMP-AMP (cGAMP) synthase (cGAS), plays a critical role in DAH under *Tank* deficiency. TANK controls cGAS-mediated cGAMP production and suppresses DNA-mediated induction of IFN-stimulated genes in macrophages by inhibiting the formation of DNA-cGAS aggregates containing ubiquitin. Collectively, TANK inhibits the cGAS-dependent recognition of cytoplasmic DNA to prevent fatal DAH in the murine lupus model.**

## Introduction

Systemic lupus erythematosus (SLE) is one of the autoimmune diseases characterized by a complex clinical syndrome comprising vasculitis, glomerulonephritis, skin rashes, nervous system symptoms, and diffuse alveolar hemorrhage (DAH) (Zamora et al, 1997). In SLE, autoantibodies such as anti-Sm and anti–double-stranded DNA (dsDNA) Abs are produced and deposited in tissues as the immune complex, causing inflammation and organ damage.

Aberrant production of cytokines is critical for the pathogenesis of SLE. Particularly, type I IFNs are strongly associated with human SLE, and the levels of type I IFNs and the expression of type I IFN regulated genes (IFN signatures) are correlated with the pathogenesis of SLE (Bancherau & Pascual, 2006). Furthermore, proinflammatory cytokines such as IL-6, IL-17, and IL-18 are also implicated in SLE (Tsokos et al, 2016). Thus, the aberrant production of type I IFNs and proinflammatory cytokines is central to the pathology of SLE. Understanding the regulatory mechanisms

of this production is of great value for the therapeutic development of SLE.

DAH is a rare but serious life-threatening complication of SLE (Zamora et al, 1997; Kazzaz et al, 2015). DAH accompanied with SLE is mainly induced by pulmonary capillaritis, which leads to the disruption of the membrane integrity of capillary walls and the leakage of blood into alveoli. The patients with DAH suffer from hypoxemia, dyspnea, cough and hemoptysis, often with concomitant infections. It is postulated that the deposition of immune complexes to the alveolar walls and pulmonary vessels contribute to the development (Santos-Ocampo et al, 2000). However, the detailed mechanisms how DAH is induced in SLE and which cytokines contribute to this pathogenesis is still unclear.

Type I IFNs and cytokines are mainly produced by innate immune cells including macrophages and DCs including plasmacytoid DCs (pDCs) (Honda et al, 2006; Takeuchi & Akira, 2010). These cells activate the transcription of type I IFN and cytokine genes after the detection of pathogen-associated molecular patterns via pattern-recognition receptors (PRRs) including TLRs, RIG-I-like receptors (RLRs), and cyclic GMP-AMP (cGAMP) synthase (cGAS), which signal through adaptor proteins, MyD88 or TRIF, MAVS, and STING (stimulator of IFN genes), respectively (Takeuchi & Akira, 2009; Kato et al, 2011; Chen et al, 2016b; Fitzgerald & Kagan, 2020; Hopfner & Hornung, 2020). Whereas these PRRs trigger distinct intracellular signaling pathways, all of the pathways converge to the activation of transcription factors IFN-regulatory factor (IRF) 3/7 as well as NF-κB, transactivating type I IFNs and proinflammatory cytokines, respectively.

The PRR signaling pathways are negatively regulated by various cellular proteins. TRAF-family member associated NF-κB activator (TANK), also known as I-TRAF, is one of such proteins suppressing the NF-κB pathway downstream of the TLR signaling by inhibiting the ubiquitination of TNF receptor–associated factor 6 (TRAF6) (Cheng & Baltimore, 1996; Rothe et al, 1996; Kawagoe et al, 2009). *Tank*-deficient mice spontaneously develop lupus-like glomerular nephritis and production of autoantibodies, which is mediated by IL-6 (Kawagoe et al, 2009). Moreover, TANK associates with TANK binding kinase 1 (TBK1) and IκB kinase-*i*/-*ε*, kinases critical for the production of type I IFN by phosphorylating IRF-3/-7 (Pomerantz & Baltimore, 1999; Fitzgerald et al, 2003; Sharma et al, 2003). Although

Department of Medical Chemistry, Graduate School of Medicine, Kyoto University, Kyoto, Japan

Correspondence: otake@mfour.med.kyoto-u.ac.jp

TANK is reported to be required for the induction of type I IFNs (Gatot et al, 2007; Guo & Cheng, 2007), production of type I IFNs in response to Newcastle disease virus (NDV), an RNA virus, was not impaired in *Tank*-deficient DCs (Kawagoe et al, 2009), indicating that TANK is dispensable for the RIG-I signaling pathway in innate immune cells. In human, an SNP in TANK is associated with SLE in Swedish cohort, suggesting that TANK is also involved in the pathogenesis of human SLE (Sandling et al, 2011). Nevertheless, it remains unknown if TANK regulates IFN responses upon the stimulation to other PRRs than RIG-I and how this regulation contributes to autoimmune pathogenesis.

In this study, we investigated the role of TANK in a pristane (2.3.-tetramethylpentadecan, TMPD)-induced lupus model, and found that TANK is essential for the prevention of fatal DAH by inhibiting lung vascular endothelial cell death. TANK deficiency resulted in the enhanced expression of type I IFNs in innate immune cells after pristane treatment, and the type I IFN signaling is essential for lethality induced by pristane treatment under TANK deficiency. The STING signaling pathway activated by intracellular dsDNA is negatively regulated by TANK in addition to the TLR-MyD88-TRAF6 pathway, and STING is critical for pristane-induced severe DAH under *Tank* deficiency. Mechanistically, TANK functions to suppress formation of dsDNA-cGAS aggregation. Together, our study revealed that TANK is critical for preventing pristane-induced fatal DAH in mice via the negative regulation of cGAS-dependent recognition of cytoplasmic DNA.

# Results

### Development of fatal DAH in $Tank^{-/-}$ mice after pristane treatment

To determine the involvement of TANK in the development of DAH, we took advantage of pristane, which induces SLE-like autoimmune disease including DAH in mice depending on type I IFN signaling (Lee et al, 2008; Reeves et al, 2009). We first examined whether pristane exacerbates the phenotypes of $Tank^{-/-}$ mice. Surprisingly, $Tank^{-/-}$ mice started to die 8 d after pristane treatment and eventually the mortality rate of pristane-treated $Tank^{-/-}$ mice increased to 89%, whereas most wild-type (WT) mice survived at 40 d after pristane injection (Fig 1A). We observed severe DAH in about 70% of $Tank^{-/-}$ mice 7 d after pristane treatment, although WT mice developed DAH much less frequently at this time point (Fig 1B). Most of WT mice caused DAH at 14 d after treatment, although eventually recovered and did not succumb to the DAH (Fig S1A). $Tank^{-/-}$ mice showed severe anemia at 7 d after pristane treatment, whereas WT mice did not decrease hemoglobin levels even at 14 d after pristane when they show DAH (Fig S1B). These results suggest that $Tank^{-/-}$, but not WT, mice caused fatal anemia due to DAH. Histological analysis revealed that pristane treatment induced vasculitis in $Tank^{-/-}$ mice, but not in WT, as evidenced by the fragmentation of leukocytes (Fig 1C) and IgM and complement C3 deposition in perivascular lesion (Fig 1D and E). Given that microvascular inflammation in the pulmonary capillary is suggested to be the cause of DAH (Lee et al, 2019), we hypothesized that vascular endothelial

cells were damaged in $Tank^{-/-}$ mice after pristane treatment. Indeed, terminal deoxynucleotidyl transferase dUTP nick end labeling (TUNEL) staining showed that pristane treatment highly increased TUNEL-positive pulmonary vascular endothelial cells in $Tank^{-/-}$ lung compared with WT at days 1 and 6 (Fig 1F–H), indicating that apoptosis of vascular endothelial cells was induced in $Tank^{-/-}$ lung in response to pristane treatment. In contrast to the development of severe DAH, pristane treatment of $Tank^{-/-}$ mice did not lead to the cause of hepatic, pancreatic, or acute renal failure as examined by the serum levels of transaminases (AST and ALT), urea nitrogen (BUN), creatinine (Cre), and albumin (Alb), or histological changes in glomeruli and heart (Fig S1C–F). These data demonstrate that pristane causes the apoptosis of pulmonary vascular epithelial cells which leads to fatal DAH in mice under TANK deficiency.

### Type I IFN signaling but not IL-6 mediates fatal DAH in $Tank^{-/-}$ mice after pristane treatment

Because TANK is involved in the regulation of the innate immune signaling pathway (Kawagoe et al, 2009), we next investigated the contribution of cytokines involved in the pristine-induced death in $Tank^{-/-}$ mice. Although IL-6 is critical for the spontaneous development of glomerular nephritis and autoantibody production in $Tank^{-/-}$ mice, IL-6 deficiency failed to improve the survival rate of $Tank^{-/-}$ mice nor the prevalence of DAH (Fig 2A and B). In sharp contrast, the abrogation of type I IFN signaling by the lack of IFN receptor (*Ifnar2*) ameliorated DAH, and dramatically rescued $Tank^{-/-}$ mice from pristane-induced death (Fig 2A and B). Thus, the type I IFN signaling, but not IL-6, is critical for the development of pristane-induced fatal DAH under TANK deficiency.

Given that TANK suppresses production of autoantibodies and natural Abs, we examined if elevated Ab production is involved in DAH under *Tank* deficiency by measuring serum anti-dsDNA Ab and total IgG1 and IgM Abs. Although lack of IL-6 decreased the production of these Abs in $Tank^{-/-}$ mice (Fig 2C and D) (Kawagoe et al, 2009), the abrogation of the type I IFN signaling by *Ifnar2* deficiency did not reduce, but rather increased the levels of Abs (Fig 2C and D). Furthermore, 6-mo-old $Tank^{-/-}Ifnar2^{-/-}$ mice developed glomerulonephritis with mesangial cell proliferation and expansion of the mesangial matrix (Fig S1G), whereas the absence of IL-6 completely prevented the mice from glomerulonephritis as previously reported (Kawagoe et al, 2009). These data demonstrate that pristane-induced lethality of $Tank^{-/-}$ mice requires the type I IFN signaling, but not IL-6, which is in contrast to the requirement of these cytokines to the development of autoimmunity under *Tank* deficiency.

### TANK suppresses the recruitment of innate immune cells to peritoneal cavity and IFN induction

Then we investigated the cell type(s) producing type I IFNs in pristane-treated $Tank^{-/-}$ mice. We have previously demonstrated that CD11b⁺Ly6C$^{high}$ cells (Ly6C$^{high}$ monocytes) are recruited to peritoneal cavity after intraperitoneal injection of pristane, and Ly6C$^{high}$ monocytes are the major source of type I IFN which is critical for autoimmunity in WT mice (Lee et al, 2008). First, intraperitoneal pristane treatment recruited slightly higher numbers of peritoneal exudate cells (PECs) in $Tank^{-/-}$ mice compared with WT mice (Fig 3A).

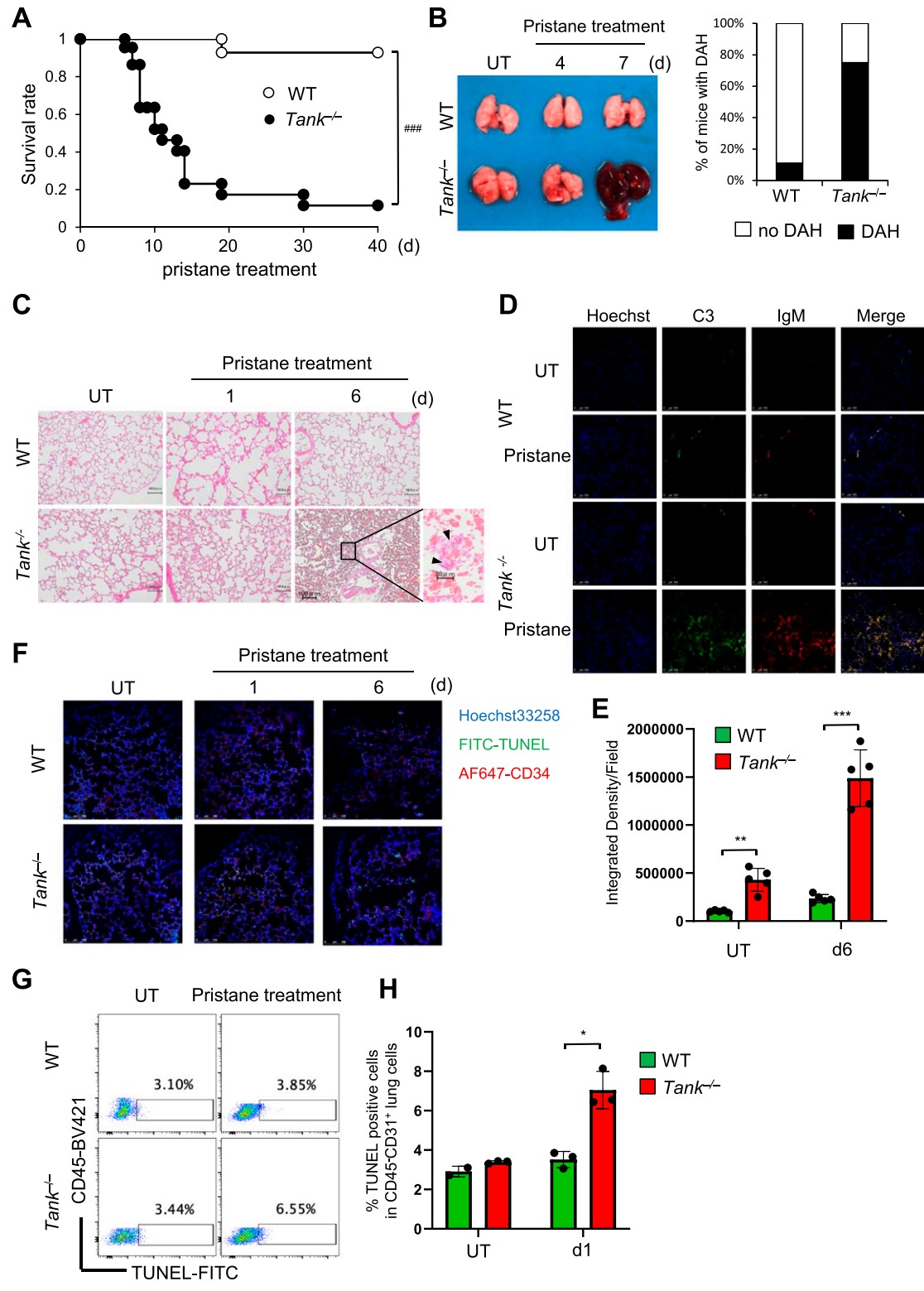

**Figure 1.   Development of fatal diffuse alveolar hemorrhage in pristane-treated *Tank*⁻/⁻ mice.**
**(A)** Survival rate of WT (n = 16, 8 males and 8 females) and *Tank*⁻/⁻ (n = 20, 10 males and 10 females) mice observed daily after a single i.p. administration of pristane. ###*P* < 0.001, (Log-rank test). **(B)** Prevalence of diffuse alveolar hemorrhage according to gross pathology in the lungs of WT (n = 12) and *Tank*⁻/⁻ mice (n = 12) harvested 7 d after pristane treatment. **(C, D, E)** Lung sections from WT and *Tank*⁻/⁻ mice at indicated time point after pristane treatment were stained with H&E (C) or with FITC-labeled anti-mouse C3 and Rhodamin-X labeled IgM (D). Scale bars represent 100 μm except a high magnification image of *Tank*⁻/⁻ lung 6 d after pristane treatment whose scale bar is 20 μm. **(E)** Integrated densities of C3 and IgM merged images were quantified shown in (E). Scale bars represent 100 μm. **(F)** TUNEL staining of CD34⁺ vascular endothelial

FACS analysis revealed that the proportion and the number of Ly6C$^{high}$ monocytes were increased in $Tank^{-/-}$ mice compared with WT, whereas the number as well as proportion of CD11b$^+$Ly6G$^{high}$ neutrophils was comparable between $Tank^{-/-}$ and WT mice (Fig 3B and C). pDCs are also known to produce large amounts of type I IFN upon viral infection (Bancherau & Pascual, 2006). Pristane-induced pDC number was also increased in $Tank^{-/-}$ mice compared with WT mice at 1 d after pristane treatment (Fig 3C). Because pristane treatment induces type I IFNs in myeloid cells recruited to the peritoneal cavity (Lee et al, 2008), we next examined the activation of IFN signatures in PECs after pristane treatment. Interestingly, the expression level of $Ifnb1$ and IFN inducible genes (ISGs), such as $Isg15$ and $Cxcl10$ are higher in $Tank^{-/-}$ PECs than WT (Fig 3D). These findings indicate that TANK suppresses pristane-induced recruitment of Ly6C$^{high}$ monocytes and pDCs to peritoneal cavity and increased the production of type I IFNs.

### TANK inhibits type I IFN responses mediated by TLR and cGAS signaling

These results prompted us to investigate the molecular mechanisms how TANK controls pristane-mediated expression of type I IFNs in innate immune cells. The TLR7-MyD88 pathway is known to contribute to the production of type I IFNs in monocytes after pristane treatment in vivo (Lee et al, 2008). Although TLR7 does not directly recognize pristane, the responsiveness against TLR7 ligands was augmented by the treatment of cells with pristane. RNA molecules from dying cells are implicated as the ligands for TLR7 (Lee et al, 2008). Indeed, TANK expression suppressed the IRF7-induced IFN-$\beta$ promoter activity which is induced downstream of MyD88 and TRAF6 (Fig 4A). Reciprocally, $Tank$-deficient BM pDCs showed much higher expression of $Ifnb1$ and $Irf7$ in response to TLR7 and TLR9 ligands (Fig 4B).

Besides TLRs, cytoplasmic nucleic acid sensors, RIG-I-like receptors (RLRs) and cGAS, induce production of type I IFNs via MAVS- and STING-dependent signaling pathways, respectively (Kato et al, 2011; Chen et al, 2016b; Hopfner & Hornung, 2020). Consistent with our previous report (Kawagoe et al, 2009), the induction of ISGs against transfection of Poly (I:C), a dsRNA analogue activating RLRs, was not elevated in the absence of TANK in conventional BMDCs (Fig 4C). We next investigated if TANK is potent to modify responses against introduction of cytoplasmic dsDNAs. To our surprise, $Tank^{-/-}$ DCs and macrophages expressed higher amount of ISGs including $Ifnb1$, $Isg15$, and $Cxcl10$ in response to dsDNA transfection compared with WT cells (Fig 4C). These results demonstrate that TANK suppresses type I IFN responses induced by TLR and cGAS, but not RLR, signaling.

### STING signaling is critical for pristane-induced lethality in $Tank^{-/-}$ mice

Then, we examined the contribution of TLR, RLR, and cGAS signaling pathways in pristane-induced lethality under $Tank$ deficiency. We found that the deletion of TLR7 or MyD88 resulted in the modest improvement of the survival in $Tank^{-/-}$ mice after pristane treatment (Fig 5A), suggesting that the signaling pathway(s) other than TLR7 is responsible for the pathogenesis. Moreover, the involvement of the RLR-MAVS pathway is also quite modest, which is consistent with the aforementioned data (Fig 5A).

In sharp contrast, the deficiency of STING ($Tmem173$), the downstream molecule of cGAS signaling, greatly reduced pristane-induced mortality on $Tank^{-/-}$ mice (Fig 5A). The augmentation of peritoneal Ly6C$^{high}$ monocyte recruitment in response to pristane under $Tank$ deficiency was ameliorated by the co-deletion of STING (Fig 5B and C). On the other hand, the recruitment of neutrophils and pDCs was only modestly affected by the absence of TANK and STING (Fig 5B), suggesting that STING specifically affect the recruitment of Ly6C$^{high}$ monocytes in $Tank^{-/-}$ mice. Furthermore, $Tmem173$ deficiency suppressed pristane-induced expression of $Ifnb1$, $Isg15$, and $Cxcl10$ in $Tank^{-/-}$ PECs (Fig 5D). These results clearly demonstrate that exacerbation of pristane-induced lethality and the production of type I IFNs in $Tank^{-/-}$ mice depends on the pathway mediated by STING.

### TANK suppresses viral and endogenous dsDNA-induced type I IFN responses

Then we examined if TANK regulates IFN responses induced by the recognition of exogenous and endogenous dsDNAs via cGAS. First, the expression of ISGs against Vaccinia virus (VACV), a DNA virus, but not to NDV, an RNA virus, was augmented in $Tank^{-/-}$ GM-CSF-induced DCs (Fig 6A). The results indicate that TANK is potent to suppress induction of type I IFNs in response to DNA, but not RNA, virus infection. In the case of pristane-induced IFN responses in monocytes, the cGAS-STING pathway is supposed to be activated by endogenous DNA, which can be released from mitochondria or from extranuclear chromatin forming micronuclei generated due to genotoxic stress (Rongvaux et al, 2014; White et al, 2014; Mackenzie et al, 2017). Consistent with this notion, $Tank^{-/-}$ macrophages shows elevated expression of $Ifnb$ compared with WT cells in response to mitochondrial DNA induced by ABT737 (pan Bcl-2 inhibitor) and Z-VAD-Fmk (caspase inhibitor) treatment (Fig 6B). Collectively, the results demonstrate that TANK suppresses type I IFN responses induced by both exogenous and endogenous cGAS ligands.

When we checked the activation of TBK1, a kinase activated downstream of STING, $Tank^{-/-}$ PECs exhibited elevated phosphorylation of TBK1 as well as IRF3 in response to Herring testis DNA stimulation compared with WT cells (Fig 6C). Thus, TANK restricts dsDNA-induced IFN reactions by suppressing signaling pathways upstream of the TBK1 phosphorylation.

### TANK may inhibit generation of DNA-cGAS aggregates harboring ubiquitination

We then investigated the mechanisms how TANK specifically suppresses dsDNA-mediated IFN responses. We initially hypothesized that

cells in the lung at indicated time point after pristane treatment. Scale bars represent 100 $\mu$m. **(G, H)** Representative FACS analysis of TUNEL-stained lung cells with or without pristane treatment (G). **(H)** TUNEL-positive cells in CD45$^-$CD31$^+$ lung cells were quantified and shown in (H).
Source data are available for this figure.

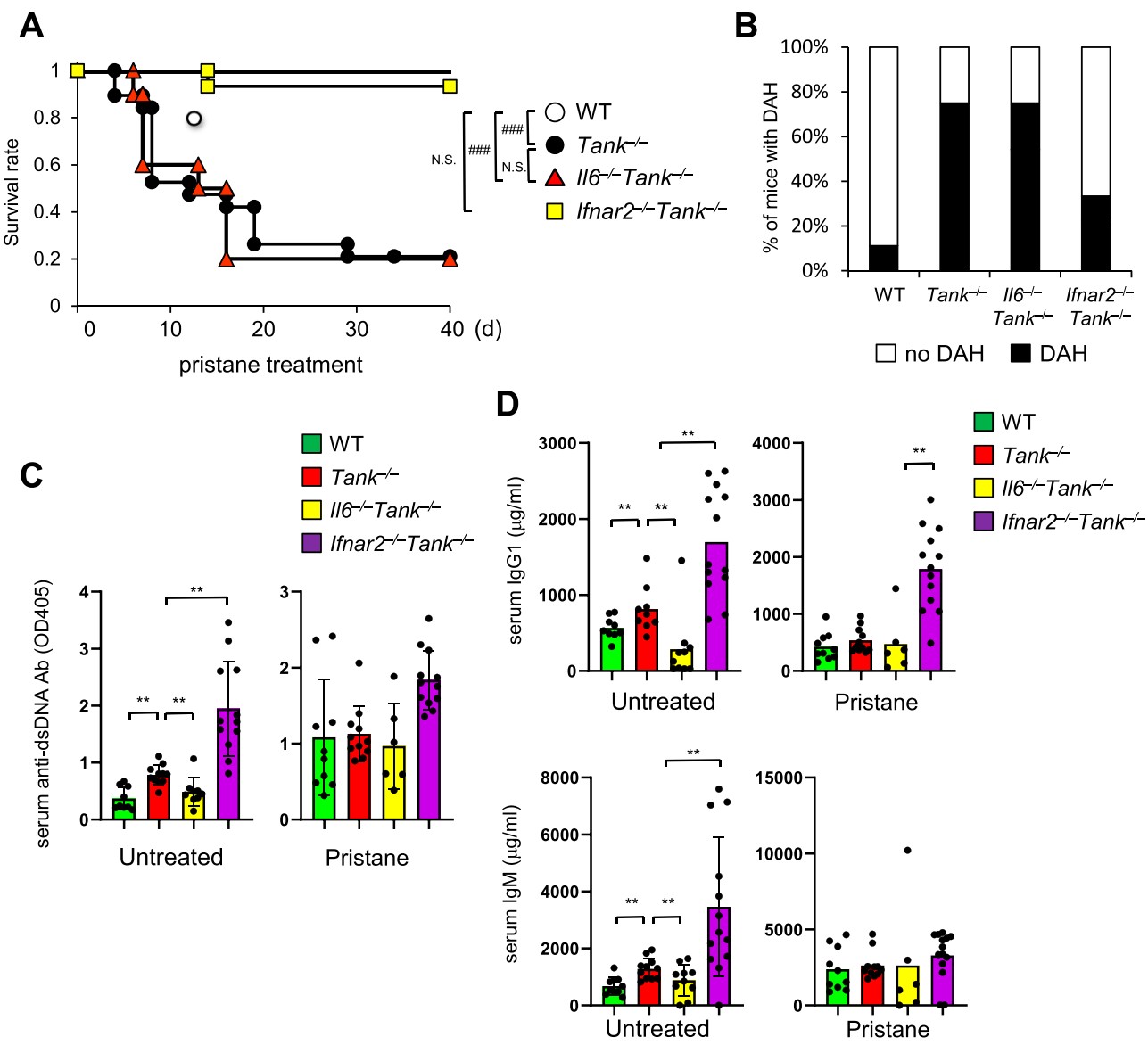

**Figure 2. Critical role of the IFN signaling in pristane-induced fatal diffuse alveolar hemorrhage in *Tank*[−/−] mice.**
**(A)** Survival rate of WT (n = 10, 6 males and 4 females), *Tank*[−/−] (n = 19, 10 males and 9 females), *Il6*[−/−]*Tank*[−/−] (n = 10, 5 males and 5 females) and *Ifnar2*[−/−]*Tank*[−/−] (n = 15, 7 males and 8 females) mice observed daily following a single i.p. administration of pristane. ###*P* < 0.001, N.S., not significant (Log-rank test). **(B)** Prevalence of diffuse alveolar hemorrhage according to gross pathology in the lungs of WT (n = 9), *Tank*[−/−] (n = 12), *Il6*[−/−]*Tank*[−/−] (n = 13), and *Ifnar2*[−/−]*Tank*[−/−] (n = 10) mice harvested 7 d after pristane treatment. **(C, D)** Serum anti–double-stranded DNA Ab levels (C) and total IgG1 or IgM levels (D) in WT (n = 10), *Tank*[−/−] (n = 11), *Il6*[−/−]*Tank*[−/−] (n = 10), and *Ifnar2*[−/−]*Tank*[−/−] (n = 12) mice with or without 7 d pristane treatment. Results are representative of at least three independent experiments. *P* < 0.05, **P* < 0.01, ***P* < 0.001, N.S., not significant (*t* test).
Source data are available for this figure.

TANK inhibits the activation of TBK1 by suppressing signaling of STING, which interacts with cyclic GMP-AMP (cGAMP), a second messenger generated by cGAS via the recognition of dsDNA (Chen et al, 2016b; Hopfner & Hornung, 2020). However, cGAMP stimulation-induced ISG expression was comparable between WT and *Tank*-deficient macrophages, whereas dsDNA stimulation induced more type I IFN and ISGs in *Tank*[−/−] macrophages (Fig 6D). In addition, stimulation with DMXAA (5,6-dimethyl-9-oxo-9H-xanthene-4-acetic acid), a murine STING agonist (Prantner et al, 2012; Conlon et al, 2013), resulted in the comparable expression of ISGs between WT and *Tank*[−/−] macrophages

(Fig 6E). Thus, TANK is suggested to suppress the IFN responses against dsDNA stimulation upstream of the STING activation.

Consistently, the production of cGAMP in response to dsDNA stimulation was increased in *Tank*[−/−] macrophages compared with WT cells (Fig 7A). After transfection of cells with dsDNA, cGAS, and dsDNA form puncta which represent cGAS undergoing a liquid-like phase transition (Du & Chen, 2018; Hopfner & Hornung, 2020). Interestingly, although the expression of cGAS was comparable between WT and *Tank*[−/−] macrophages (Fig 7B), the numbers of cGAS-DNA puncta were significantly increased in *Tank*[−/−] macrophages

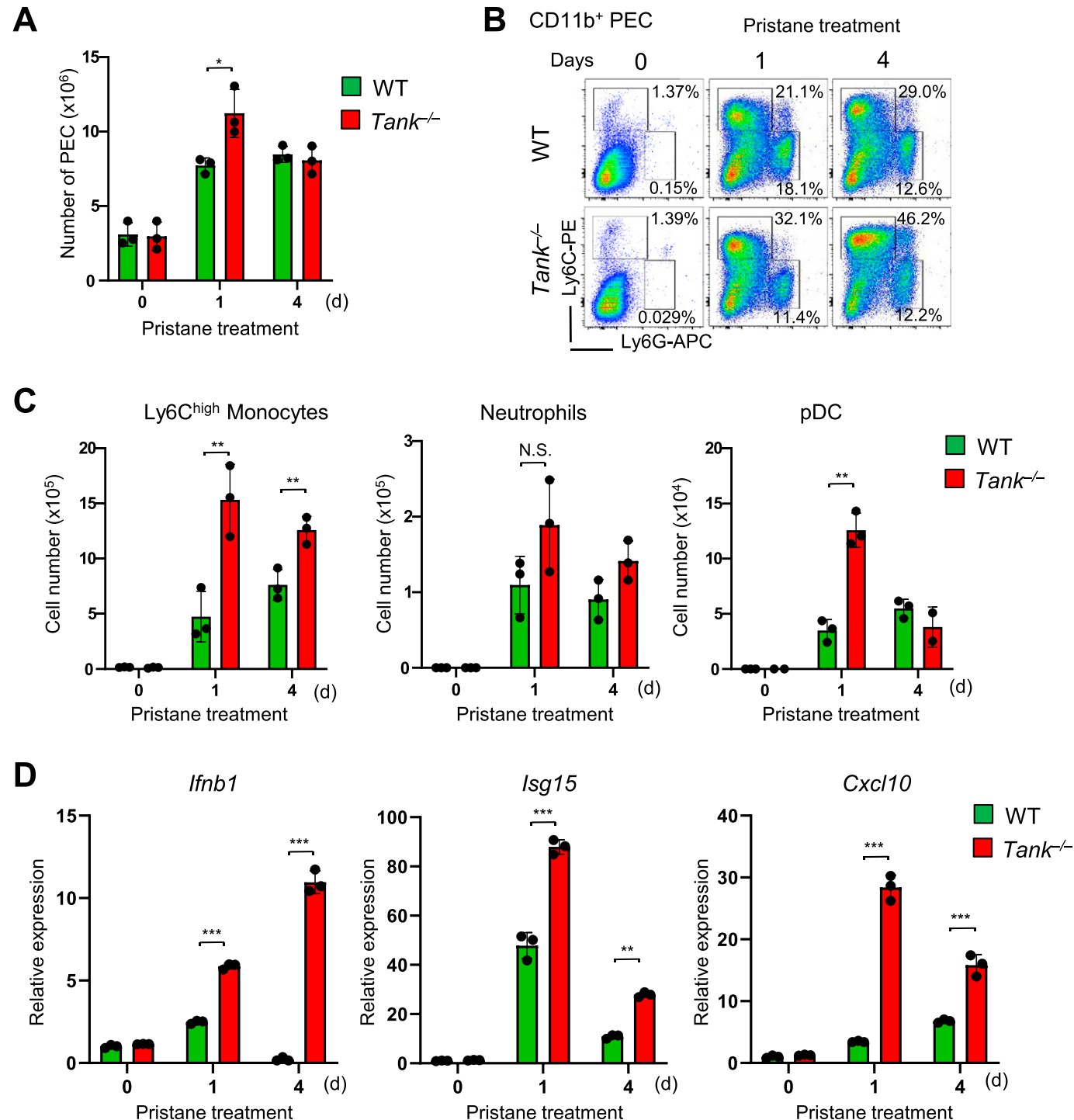

**Figure 3. TANK suppresses recruitment of innate immune cells to peritoneal cavity and ISG induction.**
**(A)** Total numbers of peritoneal exudate cells (PECs) after pristane treatment in WT (n = 3) and $Tank^{-/-}$ (n = 3) mice. **(B)** Flow cytometry analysis of PECs in WT and $Tank^{-/-}$ mice. The proportion of Ly6C$^{high}$ inflammatory monocytes and Ly6G$^+$Ly6C$^{int}$ neutrophils among CD11b$^+$ PECs is shown. **(C)** Numbers of Ly6C$^{high}$ monocytes, Ly6G$^+$Ly6C$^{int}$ neutrophils, and CD11c$^+$PDCA1$^+$ pDCs in PECs from WT (n = 3) and $Tank^{-/-}$ (n = 3) mice at indicated time point after pristane treatment. **(D)** Quantitative PCR analysis for the expression of $Ifnb1$, $Isg15$, and $Cxcl10$ in total PECs obtained from WT (n = 3) and $Tank^{-/-}$ (n = 3) mice with or without pristane treatment. $*P < 0.05$, $**P < 0.01$, $***P < 0.001$, N.S., not significant ($t$ test).
Source data are available for this figure.

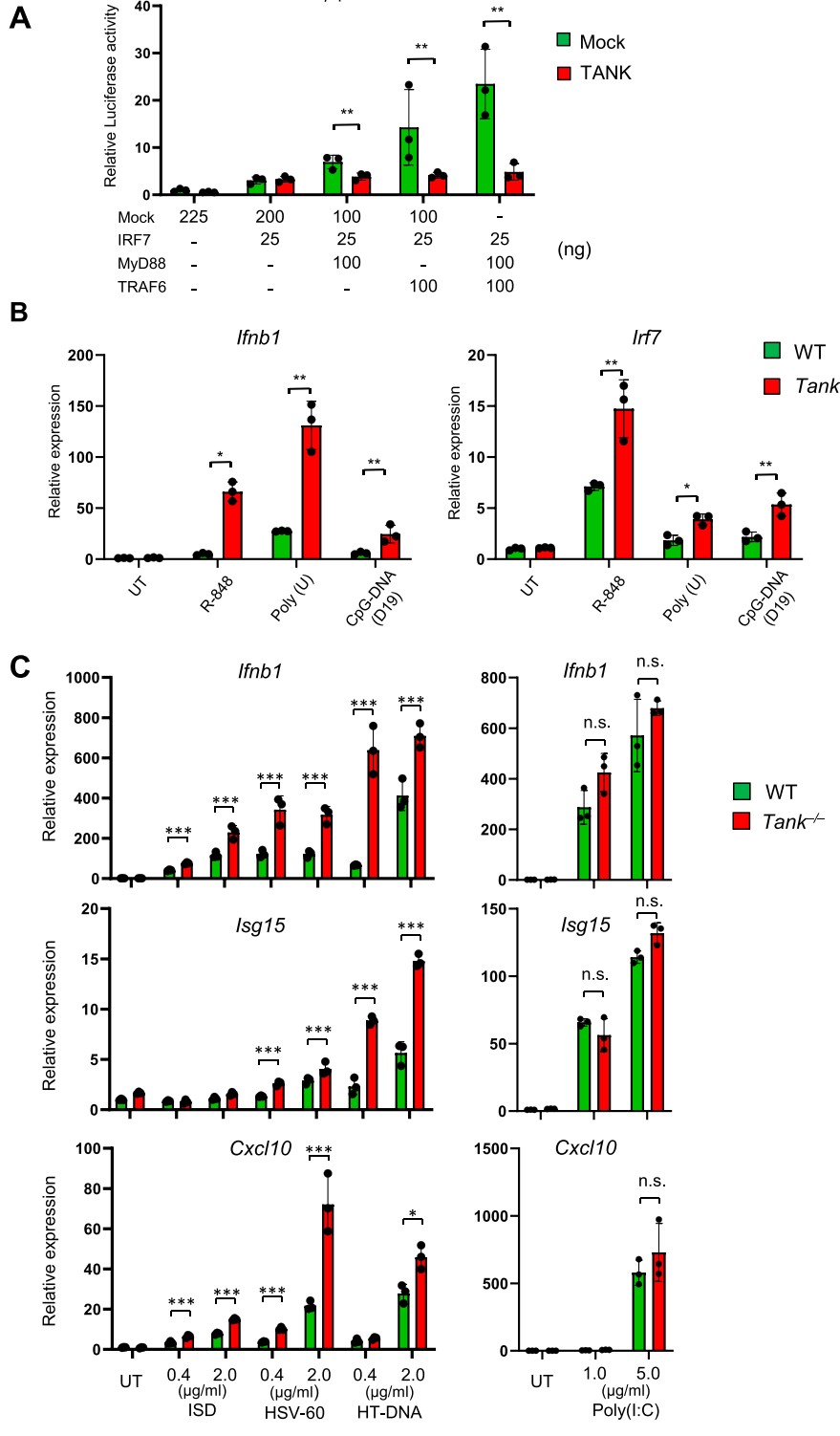

**Figure 4. TANK suppresses IFN responses induced by TLR and cGAS ligands.**
**(A)** HEK293T cells were transfected with the IFN-β-promoter luciferase reporter and Renilla control plasmids together with indicated expression plasmids with or without the TANK expression plasmid. The luciferase reporter activity was determined 48 h after transfection. **(B)** BM-plasmacytoid dendritic cell from WT and *Tank*⁻/⁻ mice were stimulated with 0.3 μM poly (U), 1 μM R-848, and 1 μM CpG-DNA for 6 h. Then, total RNA was prepared and the expression of *Ifnb* and *Irf7* was determined by QPCR. Results are representative of at least three independent experiments. **(C)** Peritoneal exudate cells from WT and *Tank*⁻/⁻ mice were transfected with indicated amounts of double-stranded DNA and poly (I:C). Total RNA was prepared 6 h after transfection, and the expression of *Ifnb1*, *Isg15*, and *Cxcl10* was examined by QPCR. *P < 0.05, **P < 0.01, ***P < 0.001 (t test). Source data are available for this figure.

than WT (Fig 7C and D). Nevertheless, TANK failed to coprecipitate cGAS in HEK293 cells even when expressed together with STING and TBK1 (Fig S2A). Reciprocally, cGAS did not coprecipitate TANK even in response to dsDNA stimulation (Fig S2B). TANK suppresses TRAF6 ubiquitination in the TLR signaling, and cGAS was reported to be

positively and negatively regulated by its ubiquitination (Wang et al, 2017; Liu et al, 2018; Seo et al, 2018; Guo et al, 2019; Wu & Li, 2020). Indeed, the cGAS-DNA aggregates are co-stained with ubiquitin both in WT and *Tank*⁻/⁻ macrophages (Fig 7E), and the numbers of cGAS-ubiquitin aggregates were significantly increased in *Tank*⁻/⁻

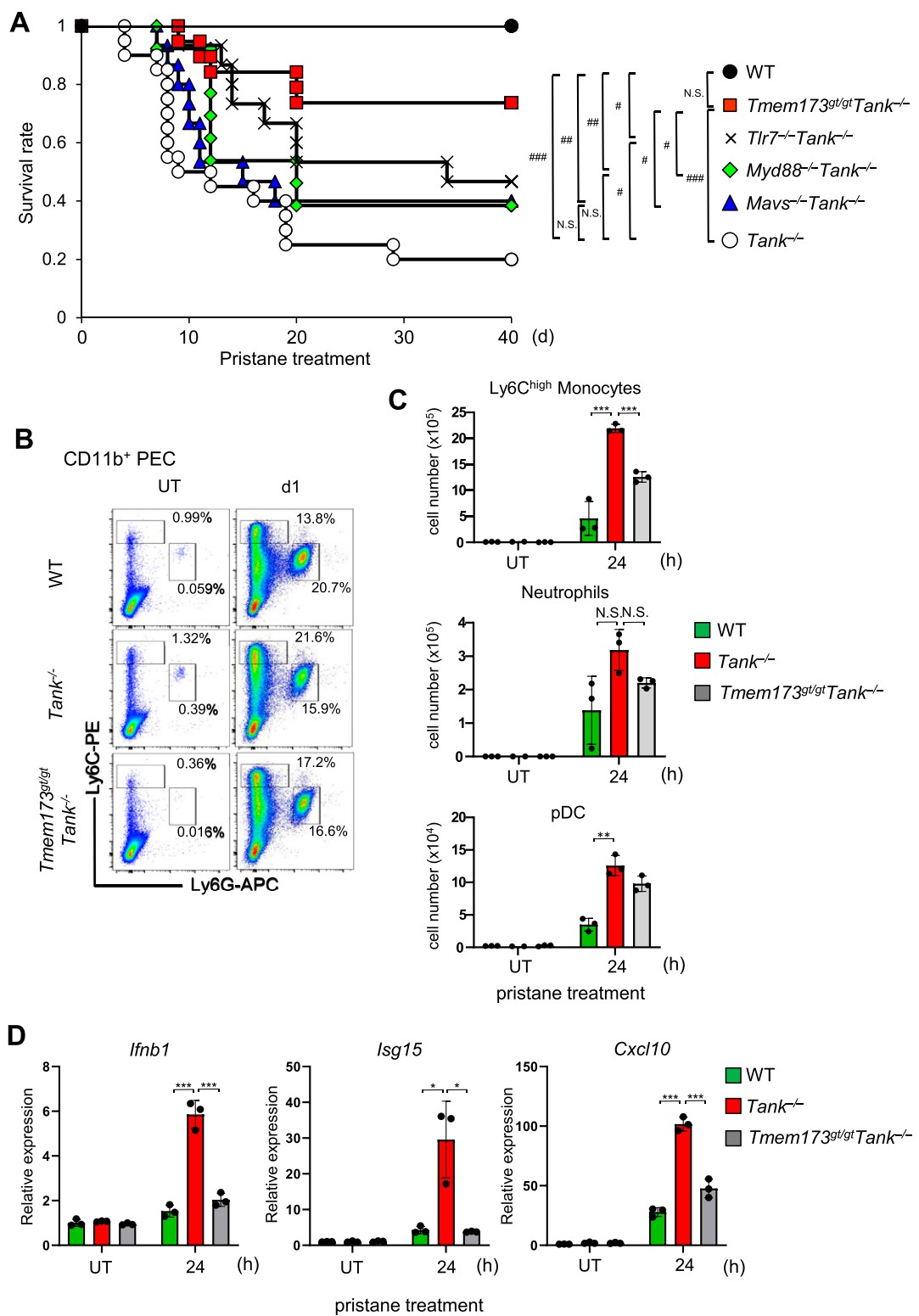

**Figure 5. STING contributes to pristane-induced lethality in *Tank*⁻/⁻ mice.**
(A) Survival rate of WT (n = 10, 6 males and 4 females), *Tank*⁻/⁻ (n = 20, 11 males and 9 females), *Tlr7*⁻/⁻*Tank*⁻/⁻ (n = 15, 8 males and 7 females), *Myd88*⁻/⁻*Tank*⁻/⁻ (n = 13, 6 males and 7 females), *Mavs*⁻/⁻*Tank*⁻/⁻ (n = 15, 7 males and 8 females) and *Tmem173*^gt/gt*Tank*⁻/⁻ (n = 19, 10 males and 9 females) mice after a single i.p. administration of pristane. #P < 0.05, ##P < 0.01, ###P < 0.001, N.S., not significant (Log-rank test). (B) Flow cytometry analysis of peritoneal exudate cells (PECs) in WT, *Tank*⁻/⁻ and *Tmem173*^gt/gt*Tank*⁻/⁻ mice. The proportion of Ly6C^high inflammatory monocytes and Ly6G⁺Ly6C^int neutrophils among CD11b⁺ PECs is shown. (C) Numbers of Ly6C^high monocytes, Ly6G⁻Ly6C^int neutrophils and CD11c⁺PDCA1⁺ plasmacytoid dendritic cells in PECs from WT (n = 3), *Tank*⁻/⁻ (n = 3) and *Tmem173*^gt/gt*Tank*⁻/⁻ (n = 3) mice at indicated time point

macrophages compared with WT cells (Fig 7E and F). These results suggest that TANK contributes to the restriction of cGAS-ubiquitin aggregates in macrophages.

# Discussion

In the present study, we demonstrate that TANK is critical for the prevention of severe DAH caused by pristane treatment, the experimental SLE model, in mice. Treatment of *Tank*-deficient mice with pristane-induced massive vascular epithelial cell death in the lung depending on the signaling through type I IFNs, but not IL-6. TANK deficiency resulted in the enhanced recruitment and expression of type I IFNs in inflammatory monocytes and pDCs after pristane treatment. STING is required for pristane-induced fatal DAH in *Tank*$^{-/-}$ mice, and TANK functions to suppress cytoplasmic dsDNA-induced IFN responses via the inhibition of DNA-cGAS aggregates. Collectively, this study contributes to the clarification of the mechanism of DAH in SLE and a regulatory role of TANK in type I IFN production.

Besides lupus-like glomerular nephritis, DAH is caused in about 60–70% of pristane-treated mice after 2–4 wk under C57BL/6 background, although the DAH is not generally fatal in WT mice (Zhuang et al, 2017). The contribution of B cells in DAH was demonstrated by the reduced prevalence of DAH in *Igμ*$^{-/-}$ mice, although T cells are dispensable (Barker et al, 2011; Zhuang et al, 2017). Consistently, IgM and C3 are known to be required for DAH (Zhuang et al, 2017). In addition, Mac-1–mediated conversion of macrophages toward classically activated macrophages promotes DAH induced by pristane (Shi et al, 2014). Indeed, a recent single cell sequencing study of lung immune cells in DAH revealed the influx of myeloid cells to the lung and inflammatory monocytes play central roles in DAH (Lee et al, 2019). Because myeloid cells such as inflammatory monocytes express type I IFNs in response to pristane, TANK expressed in inflammatory monocytes is likely to contribute to the prevention of DAH via controlling type I IFN production as well as the recruitment of inflammatory monocytes.

Among cytokines, IL-10 was reported to be important for the prevention of DAH (Zhuang et al, 2017). On the other hand, TLR and type I IFN signaling pathways are dispensable for the development of DAH in WT mice (Lee et al, 2008; Barker et al, 2011; Zhuang et al, 2017). Nevertheless, in the absence of *Tank*, the type I IFN signaling was essential for DAH pathogenesis. These observations suggest that the type I IFN responses are also involved in the pathogenesis of DAH, when levels of type I IFNs and the signaling were increased by TANK deficiency. Nevertheless, further studies are required to uncover the mechanism how increased type I IFNs contribute to the development of DAH. It is well known that type I IFN production is correlated with the pathogenesis of human SLE (Banchereau & Pascual, 2006). Given that DAH is one of severe complication of SLE,

it is possible that type IFNs also contribute to the development of DAH in human SLE patients.

In this study, we found that the cGAS-STING pathway contributes to the DAH in pristane-treated *Tank*$^{-/-}$ mice. By using *Trex1*$^{-/-}$ mice, a mouse model of human Aicardi-Goutieres Syndrome and SLE, the importance of the cGAS signaling in autoimmunity has been well demonstrated (Gao et al, 2015). Similarly, the cGAS-STING pathway is activated by the mutations of RNase H2 found in Aicardi-Goutieres Syndrome patients (Mackenzie et al, 2016; Pokatayev et al, 2016). Therefore, the cGAS-STING pathway might be suppressed by TANK even in human SLE patients to prevent pathogenesis of autoimmunity and the development of DAH. The open question is the source of endogenous DNA recognized by cGAS contributing to the cause of DAH. Self-DNAs from damaged cells after pristane treatment potentially activate cGAS, and indeed we found that the type I IFNs induced by mitochondrial DNA from damaged cells are suppressed by TANK. Interestingly, it was reported that apoptosis-derived membrane vesicles from SLE sera activates the cGAS-STING pathway to induce type I IFNs in human SLE patients (Kato et al, 2018). It is interesting to further explore if TANK inhibits endogenous DNA-mediated immune responses in human SLE patients.

We found that pristane-induced recruitment of Ly6C$^{high}$ monocytes in *Tank*$^{-/-}$ mice depends on the presence of STING. It was reported that the type I IFN signaling induced production of chemokines such as CCL2, CCL7, and CCL12, which recruits Ly6C$^{high}$ monocytes via the interaction with CCR2 on the cells (Lee et al, 2009). Thus, the STING pathway can contribute to the recruitment of Ly6C$^{high}$ monocytes via the production of chemokines activating CCR2 through the production of type I IFNs. Another unanswered question is what cell type(s) are directly activated by endogenous DNA via the cGAS-STING pathway in response to pristane treatment. In addition to Ly6C$^{high}$ monocytes, pDCs and neutrophils are also recruited to peritoneal cavity in response to pristane treatment. Future analysis of mice lacking STING in specific cell types will clarify the cells initially activated by endogenous DNA via STING pathway in response to pristane treatment.

TANK is known to interact with TBK1, and TANK is reported to be required for the activation of type I IFN induction. In contrast, this study, together with a previous report (Kawagoe et al, 2009), clearly demonstrates that TANK is not a positive regulator for the induction of type I IFN in response to various PRRs in innate immune cells. To the contrary, TANK deficiency led to the elevation of type I IFN induction in response to transfection of dsDNAs, but not to dsRNAs. Surprisingly, TANK does not directly control the activation of TBK1 or STING, suggesting that TANK suppresses the dsDNA-mediated signaling upstream of STING. Cytoplasmic dsDNA induces multimerization of cGAS culminating to the liquid–liquid phase separation forming puncta of DNA-cGAS in the cells (Du & Chen, 2018). The cGAS catalytic activity to generate cGAMP is known to be controlled by posttranslational modifications including ubiquitination (Wu & Li, 2020).

---

after pristane treatment. **(D)** Quantitative PCR analysis for the expression of *Ifnb1*, *Isg15*, and *Cxcl10* in total PECs obtained from WT (n = 3), *Tank*$^{-/-}$ (n = 3), and *Tmem173*$^{gt/gt}$*Tank*$^{-/-}$ (n = 3) mice with or without pristane treatment. *P < 0.05, **P < 0.01, ***P < 0.001, N.S., not significant (t test). Source data are available for this figure.

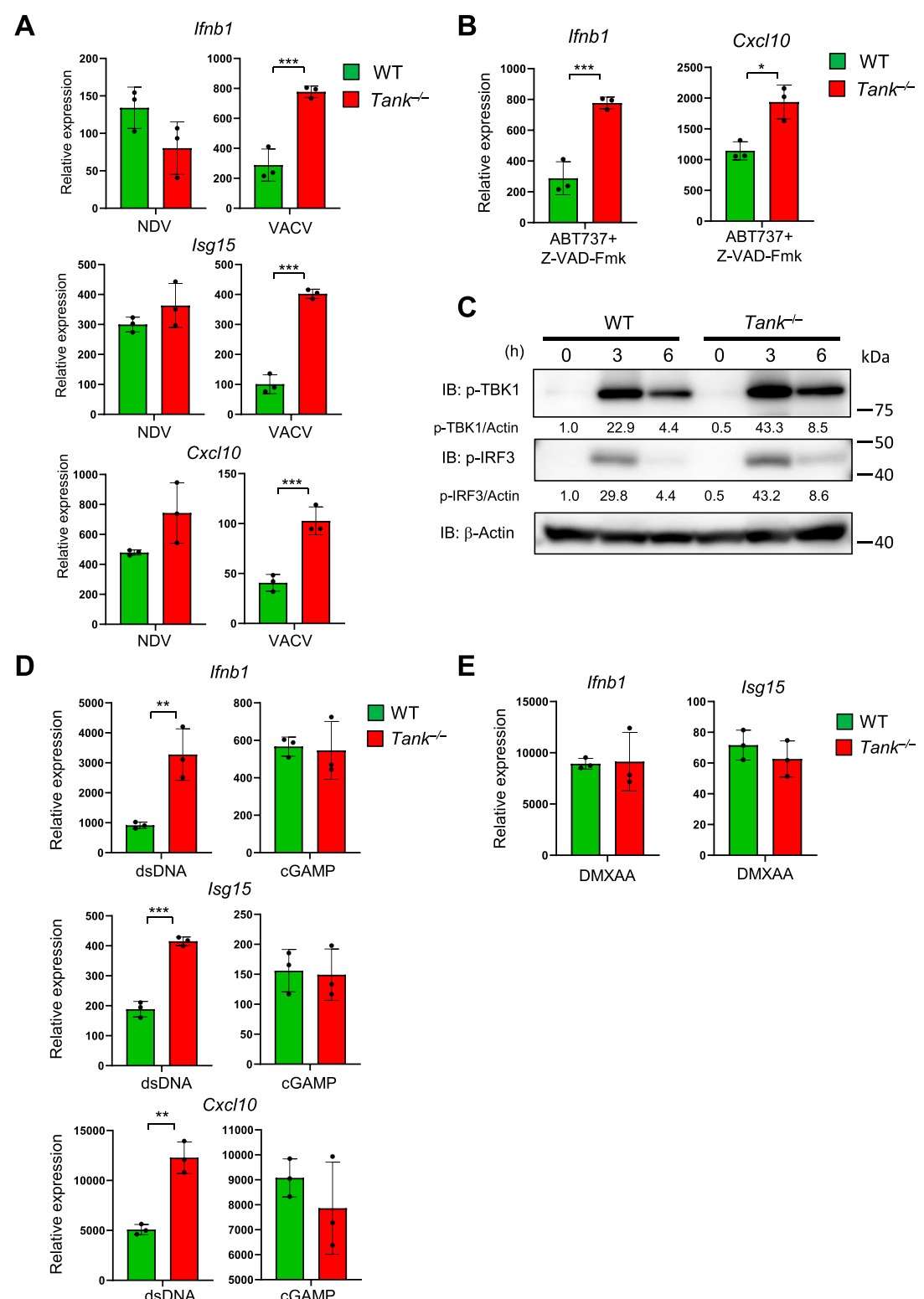

**Figure 6. TANK negatively regulates IFN responses induced by viral and endogenous double-stranded DNAs.**
**(A)** BMDCs from WT and *Tank*[-/-] mice were infected with Newcastle disease virus and VACV for 24 h. Then total RNA was prepared, and the expression of *Ifnb1*, *Isg15*, and *Cxcl10* was examined by QPCR. **(B)** BMDCs from WT and *Tank*[-/-] mice were treated with ABT737 together with Z-VAD-Fmk for 6 h, and the expression of *Ifnb* was determined by QPCR. **(C)** Macrophages from WT and *Tank*[-/-] mice were transfected with Herring testis DNA for indicated periods. Then cell lysates were prepared and subjected to the immunoblot analysis using anti-p-TBK1, p-IRF3, and β-actin Abs. p-TBK1 and p-IRF3 were quantified and normalized to β-actin. **(D)** Macrophages from WT and *Tank*[-/-] mice were stimulated with cGAMP and double-stranded DNA for 8 h. Total RNA was prepared after stimulation, and the expression of *Ifnb1*, *Isg15*, and *Cxcl10* was examined

Although TANK did not coprecipitate cGAS in the cytoplasm, TANK deficiency augmented the numbers of DNA-cGAS aggregates with ubiquitin. cGAS is reported to be polyubiquitinated by several E3 ubiquitin ligases, RNF185, TRIM56, TRIM41, and TRIM14 (Chen et al, 2016a; Wang et al, 2017; Liu et al, 2018; Seo et al, 2018). RNF185-mediated K27-linked polyubiquitination as well as TRIM56- and TRIM41-mediated monoubiquitination promotes cGAS activation (Wang et al, 2017; Liu et al, 2018; Seo et al, 2018). On the other hand, cGAS also undergoes K48-linked polyubiquitination inducing autophagic degradation, which is inhibited by deubiquitinases USP14 and USP27X (Chen et al, 2016a; Guo et al, 2019). Considering that TANK is potent to inhibit TRAF6 K63-linked polyubiquitination, TANK may suppress ubiquitination which leads to the formation of the DNA-cGAS aggregates to inhibit cGAMP generation, although we cannot exclude the possibility that TANK suppresses cGAS signaling independent of ubiquitination. Future studies will uncover the precise mechanisms how TANK specifically controls the IFN responses against cytoplasmic dsDNA stimulation.

Besides the cGAS-STING pathway, the lack of TLR7 or MyD88 also ameliorated pristane-induce lethality in TANK-deficient mice, although the contribution of TLR7 was less than STING. TLR7 was reported to be involved in the pristane-induced type I IFN production in inflammatory monocytes (Lee et al, 2008), and TANK suppresses cytokine production downstream of TLR7 like other TLRs. Thus, not only dsDNA, but also RNA derived from damaged cells after pristane treatment seems to be involved in the development of DAH under TANK deficiency.

Depletion of the type I IFN signaling did not ameliorate, but rather exacerbated autoantibody production in *Tank*-deficient mice. Although type I IFNs are critical for autoimmunity in various mouse models and even human SLE, there are autoimmunity models which develop independent of type I IFNs. These include experimental autoimmune encephalitis, DNase II deficiency and TLR7-mediated lupus nephritis mouse models, indicating that type I IFN-independent pathways contribute to the autoimmunity at certain autoimmune conditions. Given that the NF-kB signaling pathway is also negatively regulated by TANK, enhanced activation of proinflammatory cytokines by NF-kB, but not type I IFNs, via the lack of TANK can be more important for the development of long-term autoimmunity.

DAH is a rare but serious complication of SLE. The mortality of DAH in SLE is ranged 0–62%. The mechanisms for the development of DAH in SLE are not well understood. Although there are many reports about therapies for DAH in SLE that include cyclophosphamide, plasmapheresis, extracorporeal membrane oxygenation (ECMO), rituximab, mycophenolate mofetil, recombinant factor VII, and stem cell transplantation, they are general immune suppression or salvage therapies (Kazzaz et al, 2015). Thus, the clarification of the pathology of DAH is necessary for the identification of novel therapeutic targets. Interestingly, SNPs of TANK are associated with human SLE, together with other genes related with the regulation of the type I IFN responses like *IKBKE*, *STAT1*, *IL8*, and

*TRAF6* (Sandling et al, 2011). Thus, TANK might serve as a novel therapeutic target of human SLE, especially to prevent the development of DAH. Furthermore, inhibition of the type I IFN signaling or the cGAS-STING pathway is another potential targets for the treatment of DAH, which can be initially analyzed by the use of *Tank*$^{-/-}$ mice treated with the neutralizing Ab for type I IFNs as well as the inhibition of STING by antagonists like H-151 (Haag et al, 2018).

In summary, we here demonstrate that TANK functions as a critical suppresser of pristane-induced development of fatal DAH by controlling type I IFN responses via the cGAS-STING pathway by regulating DNA-cGAS aggregate formation. Further studies will uncover the mechanisms how TANK contribute to the complex pathogenicity of DAH, and may pave the way to regulate this serious complication of SLE.

# Materials and Methods

### Mice

*Tank*$^{-/-}$ mice were generated as previously described (Kawagoe et al, 2009). *Il6*$^{-/-}$, *Ifnar2*$^{-/-}$, *Tlr7*$^{-/-}$, *Myd88*$^{-/-}$, and *Tmem173*$^{gt/gt}$ mice were as described (Kawagoe et al, 2009; Kumagai et al, 2009; Sauer et al, 2011) and mice at ages between 8 and 16 wk under C57BL/6 background were used for the analysis. The animal experiments were approved by the Committee for Animal Experiments of Graduate School of Medicine, Kyoto University. WT and indicated mutant mice received a single intraperitoneal (i.p.) injection of 0.5 ml 2,6,10,14-tetramethylpentadecane (TMPD; pristane; Sigma-Aldrich).

### Reagents

dsDNA (ISD, HSV-60) was purchased from InvivoGen. Poly (I:C) was obtained from GE Healthcare. Herring testis DNA and Poly (U) was from Sigma-Aldrich, CpG-DNA (D19), R-848, 2′3′-Cyclic GMP-AMP (cGAMP), and 5,6-dimethylxanthenone-4-acetic acid (DMXAA) were purchased from InvivoGen. ABT737 and Z-VAD-Fmk were purchased from Santa Cruz and R&D Systems, respectively. NDV was as described previously (Kumagai et al, 2007). Vaccinia virus DIE strain was kindly provided by Dr. K Ishii, National Institute of Infectious Diseases. Abs specific to phospho-TBK1, phospho-IRF3, and β-actin were purchased from Cell Signaling Technology or Santa Cruz Biotechnology. Abs for flow cytometry including anti-mouse CD4, CD31, CD62L, CD44, CD45, B220, CD138, CD11b, Ly6C, and Ly6G were purchased from BD Biosciences or BioLegend. Anti-mouse PDCA1 Ab was obtained from Miltenyi Biotec. Plasmids for the IFN-β-promoter luciferase reporter, MyD88, TRAF6, IRF7, and TANK expression are described previously (Kawai et al, 2004; Kawagoe et al, 2009).

---

by QPCR. **(E)** Macrophages from WT and *Tank*$^{-/-}$ mice were treated with DMXAA for 2 h. Total RNA was prepared 24 h after transfection, and the expression of *Ifnb1* and *Isg15* was examined by QPCR. *$P < 0.05$, **$P < 0.01$, ***$P < 0.001$ ($t$ test).
Source data are available for this figure.

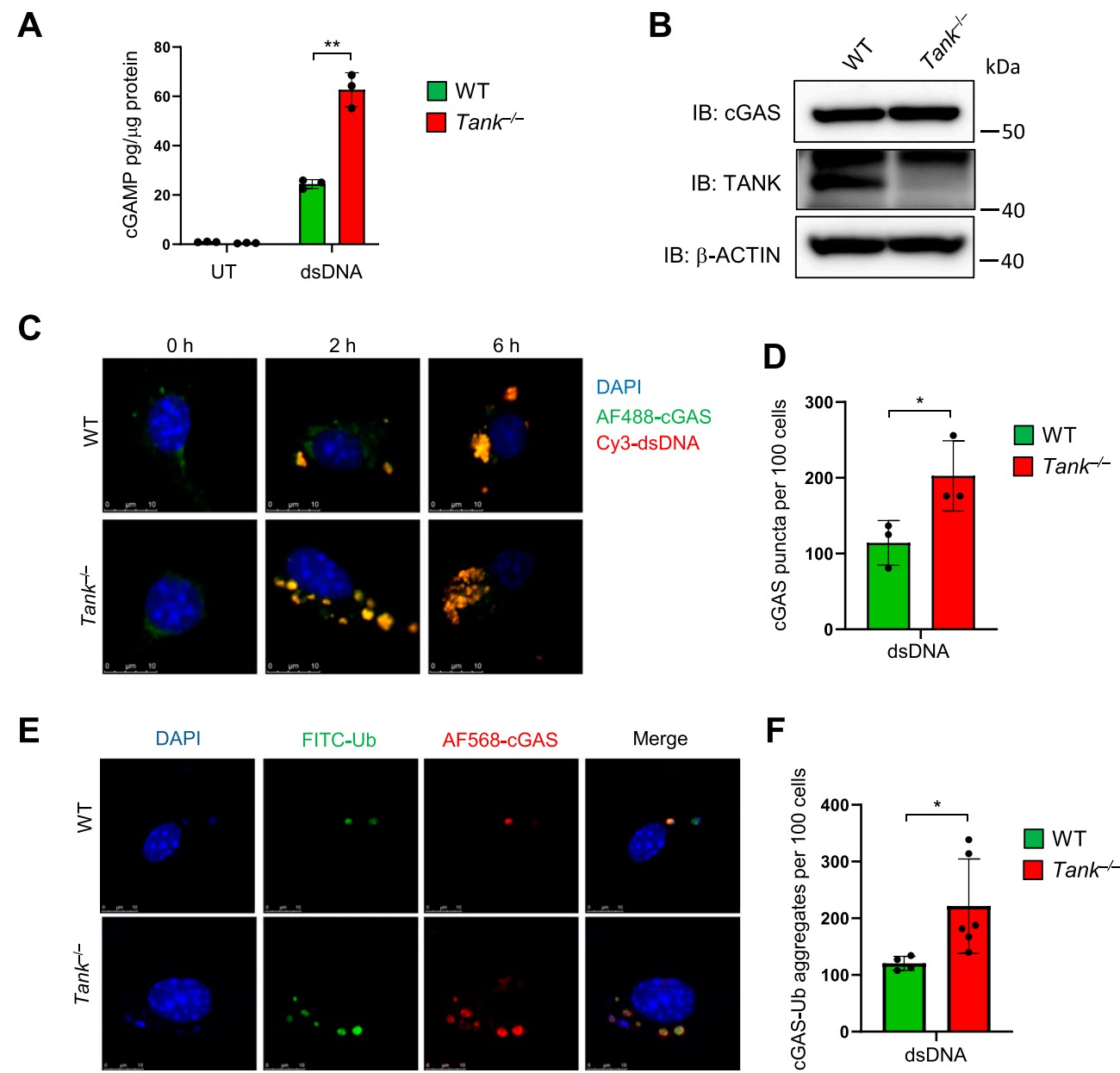

**Figure 7. TANK suppresses formation of double-stranded DNA (dsDNA)–cGAS puncta including ubiquitin.**
**(A)** Macrophages from WT and $Tank^{-/-}$ mice were stimulated with dsDNA for 2 h. Then concentrations of cGAMP in the cell lysates were measured by ELISA. **(B)** Cell lysates from WT and $Tank^{-/-}$ macrophages were subjected to the immunoblot analysis using anti-cGAS, TANK, and β-actin Abs. **(C, D)** Macrophages from WT and $Tank^{-/-}$ mice stimulated with Cy3-dsDNA for indicated periods. Then immunostaining was performed using AF488-anti-cGAS Ab and DAPI. **(C, D)** The representative images are shown in (C), and the numbers of cGAS puncta were quantified by images obtained from three independent experiments (D). **(E, F)** Macrophages from WT and $Tank^{-/-}$ mice were stimulated with dsDNA for 2 h. Then immunostaining was performed using FITC-anti-Ub (FK2) and AF568-cGAS Abs and DAPI. **(E, F)** The representative images are shown in (E), and the numbers of cGAS-Ub aggregates were quantified by images obtained from four (WT) and six ($Tank^{-/-}$) independent experiments (F). *$P < 0.05$, **$P < 0.01$ ($t$ test).
Source data are available for this figure.

## Flow cytometry

For flow cytometry analyses, PECs or splenocytes were stained with Ghost Dye Violet 510 reagent (Tonbo Biosciences) according to the manufacturer's instruction to exclude dead cells, and then treated with Ab cocktail solution containing anti-mouse CD16/CD32 (BioLegend) Ab and stained with indicated Abs. For the analysis of

TUNEL-positive cells by flow cytometry, lung single cell suspension was prepared as described previously (Nakatsuka et al, 2021), and subjected to terminal deoxynucleotidyl transferase dUTP nick end labeling (TUNEL) staining using FragEL DNA Fragmentation detection kit (Calbiochem) according to the manufacturer's instructions, together with anti-mouse CD45 and anti-mouse CD31 Abs. Data were obtained by using FACSVerse flow cytometers

(BD Biosciences) or LSRFotessa X-20 (BD Biosciences). Data were analyzed with FlowJo software (FlowJo, LLC).

## Lung pathology

Lung of pristane-treated mice were prepared at indicated time and fixed with formalin. The fixed tissues were paraffin embedded and sectioned followed by staining with hematoxylin & eosin (H&E). The development of DAH was evaluated by gross inspection of excised lung and confirmed microscopically. For the analysis of apoptotic cells, tissue sections were antigen retrieved and analyzed by TUNEL staining using FragEL DNA Fragmentation detection kit (Calbiochem) according to the manufacturer's instructions.

## Preparation of mouse cells

PECs were isolated from the peritoneal cavities of mice 3 d after injection with 2 ml of 4.0% thioglycolate medium (Sigma-Aldrich). BM cells were isolated from femurs and were cultured in RPMI 1640 medium supplemented with 10% FCS, 50 $\mu$M 2-ME, and 100 ng/ml Flt3L (BioLegend) for 7 d. Floating cells were collected with gentle agitation and used as BM-pDCs. BM cells cultured in RPMI 1640 medium supplemented with 20% FCS, 50 $\mu$M 2-ME, and 10 ng/ml GM-CSF (Peprotech) for 6 d with the replacement of culture media on day 2 and 4 were used as BMDCs. BM cells cultured in RPMI 1640 medium supplemented with 20% FCS, 50 $\mu$M 2-ME, and 20 ng/ml M-CSF (BioLegend) for 6 d were used as BM-derived macrophages. Digitonin permeabilization was used to deliver cGAMP into cultured cells as previously described (Wu et al, 2013).

## Gene expression analysis

RNA from PECs from pristane-treated mice or BMDCs were prepared using TRIzol reagent (Thermo Fisher Scientific) according to manufacturer's protocol. Then cDNA was generated with the ReverTra Ace (Toyobo). The reverse transcription reaction was subsequently used as a template for real-time PCR. Real-time PCR assays were performed on StepOnePlus (Applied Biosystems) using SYBR Green PCR master mix (Toyobo) according to the manufacturer's protocol. Data were normalized to *Actb*. The following primers were used: *Actb* forward; 5′-ATGCTCCCCGGGCTGTAT-3′, *Actb* reverse; 5′-CATAGGATCCTTCTGACCCATTC-3′, *Ifnb1* forward; 5′-CAGCTCCAA-GAAAGGACGAAC-3′, *Ifnb1* reverse; 5′-GGCAGTGTAACTCTTCTGCAT-3′, *Isg15* forward; 5′-GGTGTCCGTGACTAACTCCAT-3′, *Isg15* reverse 5′-TGGAAAGGGTAAGACCGTCCT-3′, *Cxcl10* forward; 5′-ATGCTGCCGT-CATTTTCTG-3′, *Cxcl10* reverse; 5′-ATTCTCACTGGCCCGTCAT-3′, *Irf7* forward; 5′-TGCAGTACAGCCACATACTGG-3′, *Irf7* reverse; 5′-CTCTAAACACGGTCTTGCTC-3′.

## Quantification of anti-dsDNA Ab and total IgM and IgG1 by ELISA

Serum anti-dsDNA Ab levels were determined by ELISA as described previously (Kawagoe et al, 2009). Briefly, plates were coated with 5 $\mu$g/ml calf thymus dsDNA (Sigma-Aldrich). Sera were added to the plate and further incubated with AP-conjugated anti-mouse IgG Ab after washing. Then the AP substrate (Sigma-Aldrich) was added, and absorbance was measured at 405 nm. Anti-dsDNA concentrations

were quantified according to the standard curve. Concentrations of total IgM and IgG1 levels in the sera were determined by ELISA as described previously (Kawagoe et al, 2009).

## Luciferase reporter assay

HEK293 cells on 24 well plates were transiently transfected with the 100 ng *Ifnb* promoter reporter and 20 ng control Renilla luciferase plasmids together with indicated plasmids with or without a 100 ng TANK expression plasmid. The amounts of total transfected DNA were adjusted to 445 ng/ml with a pcDNA3.1(+) empty plasmid (Mock). Cell lysates were prepared 48 h after transfection and the luciferase activity was measured by using the Dual-luciferase reporter assay system (Promega) following the manufacturer's protocol. The Renilla luciferase reporter plasmid was simultaneously transfected as an internal control.

## Immunoblot analysis

Cells were lysed in a lysis buffer containing 1% Nonidet P-40, 150 mM NaCl, 20 mM Tris–HCl (pH 7.5), 1 mM EDTA, and a protease inhibitor cocktail (Roche). Lysates were separated by SDS–PAGE and transferred onto polyvinylidene difluoride membranes (Bio-Rad). After membranes were blotted with Abs, proteins on membranes were visualized with Luminata Forte Western HRP Substrate (Millipore). Luminescence data were obtained by ImageQuant LAS 4000 (GE Healthcare). Intensities of p-TBK1 and p-IRF3 bands were quantified by using ImageJ software.

## Measurement of cGAMP concentration

Macrophages treated with dsDNA for 2 h were lysed in M-PER Mammalian Protein Extraction Reagent (Thermo Fisher Scientific) buffer and used for the measurement of cGAMP concentration by using the 2′3′-cGAMP ELISA Kit (Cayman Chemical) according to the manufacturer's instructions. Protein concentrations in the cell lysates were measured using Pierce BCA Protein Assay Kit (Thermo Fisher Scientific) and was used to normalize cGAMP concentrations.

## Immunofluorescence

Macrophages were seeded on cover slops placed on 24 well plates to 1 × 10$^5$ cells/well, and fixed with 3% paraformaldehyde in PBS for 10 min, incubated with 50 mM NH$_4$Cl in PBS for 10 min, permeabilized with 0.5% Triton X-100 for 10 min, and blocked with 2% normal goat serum (Dako) and 0.1% gelatin in PBS. Primary Abs to cGAS (D-9; Santa Cruz), FITC-conjugated mono- and poly-ubiquitinylated conjugates monoclonal Ab (FK2; Enzo) were used for staining in combination with secondary Ab conjugated to Alexa 568 goat anti-Mouse IgG (H + L) (Invitrogen). Cy3-labeled dsDNA were generated by annealing Cy3-labeled sense and anti-sense ssDNA oligos in an annealing buffer (20 mM Tris–HCl, pH 7.5, 50 mM NaCl) ramping down from 95°C to 25°C at 1°C/min, followed by NaOAc and ethanol precipitation. Annealed dsDNA oligos were resuspended into a desired buffer for experimental use. Images were captured on a TCS SPE confocal microscopes (Leica) and analyzed with the LAS-AF software (Leica). For the quantification of

cGAS and dsDNA colocalized-puncta and the quantification of integrated densities of C3 and IgM merged images, images were analyzed using ImageJ (National Institutes of Health) and the Cell Counter plugin. The ratio between puncta in WT and Tank KO BMDMs was plotted using GraphPad Prism 8.

## Statistical analysis

Statistical significance was calculated with the two-tailed *t* test or log-rank test. *P*-values of less than 0.05 were considered significant.

# Supplementary Information

# Acknowledgements

We thank Dr. Tatsuya Saitoh and all members of our laboratory for discussions, and Ms. Yoshimi Okumoto for secretary assistance. We thank Drs. Hiroki Kato and Takashi Fujita for providing mice and reagents, and Dr. Koji Ishii for providing VACV. The authors thank the Center for Anatomical, Pathological and Forensic Medical Research, Kyoto University Graduate School of Medicine, for preparing microscope slides. This research was supported by a Grant-in-Aid for Scientific Research (S) (18H05278, to O Takeuchi), AMED under Grant Number JP19gm4010002, and Japan Science and Technology Agency under grant number JPMJMS2025 (to O Takeuchi). This research was also supported by a grant from Takeda Science Foundation.

## Author Contributions

A Wakabayashi: conceptualization, data curation, formal analysis, investigation, methodology, and writing—original draft.
M Yoshinaga: investigation and methodology.
O Takeuchi: conceptualization, data curation, formal analysis, supervision, funding acquisition, methodology, project administration, and writing—original draft, review, and editing.

## Conflict of Interest Statement

The authors declare that they have no conflict of interest.

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
