## [Reviewer comments · Life Science Alliance]

Life Science Alliance

TANK prevents IFN-dependent fatal diffuse alveolar hemorrhage by suppressing DNA-cGAS aggregation

Atsuko Wakabayashi, Masanori Yoshinaga, and Osamu Takeuchi

DOI: <https://doi.org/10.26508/lsa.202101067>

Corresponding author(s): *Osamu Takeuchi, Kyoto University*

Review Timeline:

Submission Date:	2021-03-15
Editorial Decision:	2021-07-06
Revision Received:	2021-10-12
Editorial Decision:	2021-11-10
Revision Received:	2021-11-11
Accepted:	2021-11-12

Transaction Report:

July 6, 2021

Re: Life Science Alliance manuscript #LSA-2021-01067-T

Prof. Osamu Takeuchi
Kyoto University
Department of Medical Chemistry, Graduate School of Medicine
Yoshida-Konoe-cho, Sakyo-ku
Kyoto 6068501
Japan

Dear Dr. Takeuchi,

Thank you for submitting your manuscript entitled "TANK prevents IFN-dependent fatal diffuse alveolar hemorrhage by suppressing DNA-cGAS aggregation" to Life Science Alliance. The manuscript was assessed by expert reviewers, whose comments are appended to this letter. We invite you to submit a revised manuscript addressing the Reviewer comments.

Thank you for this interesting contribution to Life Science Alliance. We are looking forward to receiving your revised manuscript.

Sincerely,

B. MANUSCRIPT ORGANIZATION AND FORMATTING:

Reviewer #1 (Comments to the Authors (Required)):

The authors show evidence that diffuse alveolar hemorrhage (DAH) is more severe in Tank-deficient mice with pristane-induced lupus than in wild type controls. They suggest that DAH is a consequence of lung vascular endothelial cell death and that type I interferon (IFN) signaling and STING play a critical role. The markedly worse survival of mice lacking Tank is quite interesting, but data showing that mortality is due to DAH are somewhat meager. Also, the data supporting the involvement of endothelial cell death and STING are not that compelling.

Figure 1 shows that mortality is strikingly increased and the onset of DAH is accelerated in Tank^{-/-} mice vs. wild type. It is not clearly stated in the methods that the control mice were B6 (or that the various knockouts were on a B6 background). Assuming that is the case, the mortality and incidence of DAH are lower than expected from other reports in the literature. Generally, about 2/3 of B6 mice develop DAH by day 14 and about 1/3 succumb (vs. < 10% here). It is possible this is due to differences between animal colonies, though the previously published figures have been similar in multiple laboratories. It seems important to document that the severity of DAH is greater in Tank-deficient mice than controls or to give some other evidence that mortality is due to more severe DAH rather than to hyperinflammation from excessive NFκB activation.

The immunofluorescence shown in Figure 1 is weak. C3 and IgM staining are very weak (Fig. 1D). Critically, the signals for FITC-TUNEL and AF647-CD34 (is it actually CD31?) in Fig. 1E are very weak, making it difficult to see co-localization. In the case of Fig. 1E, it would be nice to have quantitative data in addition to the fluorescent images. Based on the data shown, it cannot be concluded that "treatment of Tank-deficient mice with pristane induces massive vascular epithelial (endothelial?) death in the lung" as stated in lines 278-280.

The flow cytometry (Fig. 1E-F, 1 mouse per condition) doesn't add much.

Figure 2A needs a survival curve for the wild type mice, not just a single data point at 14-days. Previously published data show that in wild type B6 mice, absence of the type I IFN receptor does not prevent DAH. Thus, it is unexpected that absence of the type I interferon receptor reverses the effect of knocking out Tank. The discrepancy is mentioned (lines 302-307), but a plausible explanation is not proposed.

In many instances, the sample sizes in Figures 3-8 are small (n = 3 per group). It is probably inappropriate to use Student's t-test to analyze the data with an "n" this small. Mann-Whitney may be more appropriate. There data are said to be representative of several experiments, but several repeats may not compensate for the small "n" in individual experiments. It would be useful to ask a biostatistician to advise.

It is irrelevant to look at renal pathology 7 days after pristane treatment (Fig. S1). Pristane-treated B6 mice do not develop severe glomerulonephritis to begin with, and the onset of renal disease is delayed for 4-6 months or more after pristane treatment.

Are the cells in the last column of graph Fig. 4A co-transfected with five different plasmids (TANK, IRF7, MyD88, TRAF6, and the IFN β promoter-Luc plasmid)? What were the doses of each plasmid, and what was used as controls for these multiple plasmids?

The numbers of mice in Fig. 5A are adequate, but it is again unclear why none of the wild type mice died over a period of 40 days from DAH. Nevertheless, Sting-deficiency improves survival on a Tank-deficient background. However, Tlr7-deficiency also has an effect, although less than that of Sting. Is reversal of the Tank-deficient phenotype just a function of how much interferon is being made? Or, since IL-10 is protective against DAH, could the severity of DAH be related to how much IL-10 is available?

Reviewer #2 (Comments to the Authors (Required)):

Diffuse alveolar hemorrhage (DAH) is a rare complication occurs in an autoimmune disease systemic lupus erythematosus (SLE). Mechanisms that regulate DAH development is incompletely understood. In the current manuscript, the authors focused on identifying the role and mechanism by which TANK, a negative regulator of the NF-κB signaling may inhibit DAH development. The authors used the pristane induced SLE mouse model, which developed DAH, to understand the mechanisms

that may promote DAH. They found that TANK prevents fatal DAH by negatively regulating type I interferon production via the STING pathway. Overall manuscript is significant and novel in understanding the mechanism of DAH development in SLE.

Comments:

- (1) Data presented in Figure 5A indicates that both TLR7 and cGAS-STING pathways are involved in DAH development. Pristane is likely to induce the exposure of both DNA- and RAN-based antigens. Therefore, the authors should emphasize the potential contribution of TLR7 signaling to DAH in the discussion.
- (2) Although identifying the role of TANK in regulating innate cell infiltration into peritoneal cavity is a good start, similar flow cytometry analysis should be performed for the lung to determine whether TANK is similarly controlling the innate cell infiltration into the lung, the site of DAH.
- (3) The major finding in this manuscript is that TANK prevents DNA-cGAS aggregate formation. The use of TANK^{-/-}-MyD88^{-/-} cells will help solidify TLR-independent role of TANK in DNA-cGAS aggregate formation.
- (4) Semi-quantification of data presented in Figure 1D and E will be beneficial for the readers as the images are not very clear to show co-localization. Also, there is a typo in Figure 1D- it should be Merge instead of Marge.
- (5) The authors have not discussed the potential mechanisms for increased autoantibody production in *lfnar2*^{-/-}-*Tank*^{-/-} mice.
- (6) The authors have presented representative flow plots for effector and memory CD4 T cells and plasma cells in Figure 2E and F. They should include the bar graphs (as in Figure 2C and D) of all mice analyzed.

Reviewer #3 (Comments to the Authors (Required)):

DAH is a serious complication in some SLE patients. Mechanistic insights into DAH pathology is significant and could lead to new intervention strategy. In the manuscript by Wakabayashi A et al., the authors showed that TANK is critical for preventing pristane-induced fatal DAH in mice by suppressing cGAS-STING dependent type I IFNs production. Overall, the study is novel and interesting. Data supporting the main conclusion is generally strong though some quantification will be helpful. The weakness is related to how TANK influences cGAS production of cGAMP. The current data supporting TANK regulates cGAS ubiquitination is weak. Below are my comments.

Major points:

1. In Figure 2A, 5A, the authors clearly showed that type I IFNs and STING drove DAH in TANK^{-/-} mice. The authors may want to examine if type I IFNs neutralizing antibody or STING antagonist H-151 improve survival in pristane treated TANK^{-/-} mice. Such experiments will not only strengthen their argument but also advance a new therapy for DAH.
2. SLE shows gender bias. In the methods and figure legend, please specify the sex of the mice used in the experiments.
3. Figure 1D and 1E will be benefited from some quantification. Currently, the data is not convincing.
4. It is not clear if STING expression in Ly6Chi monocyte or pDCs are responsible for type I IFNs production. Can the authors elaborate it in the Discussion? Future experiment using conditional STING knock out mice will be helpful.
5. STING affected the recruitment of Ly6Chi monocytes (Fig 3) and seems to be critical for DAH. Can the authors discuss how STING may promote monocytes infiltration?
6. In line 240, the authors stated " Tank^{-/-} PECs exhibited elevated phosphorylation of TBK1 as well as IRF3 in response to Herring testis DNA stimulation compared with WT cells (Fig. 6C). There is no quantification and the p-IRF3 and p-TBK1 looked comparable.
7. In line 273, the authors stated "Indeed, the cGAS-DNA aggregates are co-stained with ubiquitin both in WT and Tank^{-/-} macrophages (Fig. 7E), implying that TANK restricts cGAS-DNA aggregate formation by suppressing ubiquitination in the complex." I do not understand the logic of this statement. If both WT and TANK^{-/-} are co-stained with ubiquitin, how can you draw the conclusion that TANK suppresses the ubiquitination of c GAS?
8. Fig 7E is not convincing. Without quantification, it seems that TANK^{-/-} macrophage have more overall ubiquitination than the WT upon DNA stimulation. It is questionable that TANK will be the predominant factor suppressing DNA-induced ubiquitination in these macrophage.
9. Again, without quantification, in Fig 7E, the AF568-cGAS stain seemed much stronger in Tank^{-/-} cells than WT cells, which will indicate that cGAS expression increased in TANK^{-/-} macrophage compared to WT cells. This contradicted to Fig 7B showing cGAS expression is not altered in TANK^{-/-} cells. In all, the authors need substantial more data, not just a Fig 7E, to support their argument that Tank influences cGAS ubiquitination.
10. Figure S2A, the authors' interpretation of the data is not accurate. The precise interpretation for Fig S2A is "TANK does not interact with cGAS in HEK cells." The authors did not examine TANK and cGAS association in Ly6Chi monocytes or pDCs where it may influence cGAMP production.
11. In Figure S2B, there was no positive control for cGAS-HA IP. Thus, the authors can not conclude that cGAS did not pull down TANK during DNA stimulation.

Minor points:

1. Figure 2, *lfnar2*^{-/-} should be *lfnar2*^{-/-}
2. Figure 2E, the Y-axis should be CD62L.
3. Figure 2F, the CD138+ plasma cells gate should be B220dull, excluding B220 hi population.
4. Figure 4A HEK cells experiment showed that TANK inhibit IRF7, MyD88 and TRAF6 production of IFN β . This is a different

mechanism for TANK in DAH where the authors propose that TANK inhibits type I IFNs by suppressing cGAS production of cGAMP. To avoid confusion, better remove Fig 4A.

We thank the reviewers for their constructive criticisms. Below, please find our point-by-point responses to all comments and questions. We believe that we have examined each of the points raised and could respond thoroughly. Again, we would like to thank all the reviewers for being interested in our manuscript and for their thoughtful suggestions, which have definitely made this a stronger paper.

Reviewer #1 (Comments to the Authors (Required)):

The authors show evidence that diffuse alveolar hemorrhage (DAH) is more severe in Tank-deficient mice with pristane-induced lupus than in wild type controls. They suggest that DAH is a consequence of lung vascular endothelial cell death and that type I interferon (IFN) signaling and STING play a critical role. The markedly worse survival of mice lacking Tank is quite interesting, but data showing that mortality is due to DAH are somewhat meager. Also, the data supporting the involvement of endothelial cell death and STING are not that compelling.

First of all, we thank the reviewer for finding this study interesting. Regarding the concern on the cause of mortality, involvement of endothelial cell death and STING, please find our reply to the specific comments below.

Figure 1 shows that mortality is strikingly increased and the onset of DAH is accelerated in Tank^{-/-} mice vs. wild type. It is not clearly stated in the methods that the control mice were B6 (or that the various knockouts were on a B6 background). Assuming that is the case, the mortality and incidence of DAH are lower than expected from other reports in the literature. Generally, about 2/3 of B6 mice develop DAH by day 14 and about 1/3 succumb (vs. < 10% here). It is possible this is due to differences between animal colonies, though the previously published figures have been similar in multiple laboratories. It seems important to document that the severity of DAH is greater in Tank-deficient mice than controls or to give some other evidence that mortality is due to more severe DAH rather than to hyperinflammation from excessive NFKB activation.

We thank the reviewer for the constructive criticism. We used *Tank* and other knockout mice under C57BL/6 background at ages between 8 and 16 weeks, and also utilized wild-type C57BL/6 mice as the control. We added the information on the mouse background in Materials and Methods section.

As the reviewer points out, the mortality of pristane-treated wild-type (WT)

mice in this study was lower than those previous reports (Zhuang H et al. Arthritis Rheumatol. 2017 and others). Although the reason of the differences in the mortality rate is unclear, the conditions of animal housing (e.g. microbiota) might be responsible for the differences in the response against pristane. Noteworthy, although the incidence of DAH in WT mice at 14 days after pristane treatment was consistent with the previous reports as shown in Fig. S1A, they eventually recovered and most of WT mice survived.

The reviewer also suggests us to evaluate the severity of DAH in WT and *Tank*^{-/-} mice. Since severe hemorrhage can be the cause of anemia, we chronologically measured blood hemoglobin and hematocrit levels in WT and *Tank*^{-/-} mice after pristane treatment. Interestingly, *Tank*^{-/-} mice developed severe decrease in hemoglobin and hematocrit levels at 7 days after treatment and all *Tank*^{-/-} mice died before 14 days (new Fig. S1B). In contrast, WT mice did not show anemia even at 14 days after pristane treatment when DAH was observed in WT mice (new Fig. S1B). Nevertheless, we did not find hemorrhage from intestine or other organs (data not shown). Furthermore, IL-6 deficiency did not ameliorate pristane-induced lethality in *Tank*^{-/-} mice (Fig. 2A), suggesting that hyperinflammation cannot explain this phenomenon. These data demonstrate that the severe anemia observed in *Tank*^{-/-} mice, but not WT mice, is due to severe DAH. Thus, these new results suggest that the increased mortality in pristane-treated *TANK*-deficient mice is due to the development of rapid and severe DAH.

The immunofluorescence shown in Figure 1 is weak. C3 and IgM staining are very weak (Fig. 1D). Critically, the signals for FITC-TUNEL and AF647-CD34 (is it actually CD31?) in Fig. 1E are very weak, making it difficult to see co-localization. In the case of Fig. 1E, it would be nice to have quantitative data in addition to the fluorescent images. Based on the data shown, it cannot be concluded that "treatment of Tank-deficient mice with pristane induces massive vascular epithelial (endothelial?) death in the lung" as stated in lines 278-280.

We thank the reviewer for the comments. We used anti-CD34 Ab to visualize endothelial cells by immunofluorescence analysis, since both CD34 and CD31 are well recognized markers of endothelial cells.

According to the reviewer's suggestion, we stained the samples with C3 and IgM again and replaced with the new data (new Fig. 1D). We further quantified the C3 and IgM merged images by the evaluation of integrated density using ImageJ software, and the data of 5 different images are shown in new Fig. 1E. As show in the new Fig.

1D and 1E, highly augmented deposition of IgM and C3 was observed in lungs from *Tank*^{-/-} mice in response to pristane treatment.

To quantify the endothelial cell death shown in original Fig. 1E (Fig. 1F in the revised manuscript), we performed flow cytometry to quantify TUNEL positive apoptotic CD31⁺CD45⁻ endothelial cells. As shown in new Fig. 1G and 1H, the frequency of apoptotic cells in CD45⁻CD31⁺ lung cells was increased in pristane-treated mice under *Tank* deficiency.

We apologize the typo and we now corrected to “lung vascular endothelial cell death”.

The flow cytometry (Fig. 1E-F, 1 mouse per condition) doesn't add much.

We thank the reviewer for this comment. Probably the author indicates Fig. 2E and 2F, showing proportion of effector/memory CD4⁺ T cells and plasma cells. We agree that the data are not essential for this study. Thus, we have withdrawn the original Fig. 2E and 2F.

Figure 2A needs a survival curve for the wild type mice, not just a single data point at 14-days. Previously published data show that in wild type B6 mice, absence of the type I IFN receptor does not prevent DAH. Thus, it is unexpected that absence of the type I interferon receptor reverses the effect of knocking out Tank. The discrepancy is mentioned (lines 302-307), but a plausible explanation is not proposed.

The data for WT mice (n=10) shown in Figure 2A are not just a single data point, but we watched the survival daily for 40 days. Since none of the WT mice died in the experiment, we pointed out the survival data at day 40 (but not day 14). We also observed the survival of other KO mice daily. Thus, we added the explanation in the Figure legends.

As the reviewer points out, it was reported that DAH observed in WT mice was not rescued by the absence of type I IFN receptor (Zhuang H et al. Arthritis Rheumatol. 2017). However, the report did not investigate if increased type I IFN responses are involved in the pathogenesis of DAH. Thus, the results suggest that the increased type I IFNs by *Tank* deficiency contribute to the development of DAH, although the molecular mechanism is unclear. We discussed this issue as the reviewer suggests by adding the following sentences.

“These observations suggest that the type I IFN responses are also involved in the pathogenesis DAH, when levels of type I IFNs and the signaling were increased by

TANK deficiency. Nevertheless, further studies are required to uncover the mechanism how increased type I IFNs contribute to the development of DAH.”

In many instances, the sample sizes in Figures 3-8 are small (n = 3 per group). It is probably inappropriate to use Student's t-test to analyze the data with an "n" this small. Mann-Whitney may be more appropriate. These data are said to be representative of several experiments, but several repeats may not compensate for the small "n" in individual experiments. It would be useful to ask a biostatistician to advise.

We thank the reviewer for the comments on the statistical analysis. According to the reviewer's suggestion, we consulted with a specialist in the biostatistics, Dr. Alexis Vandebon at Kyoto University. His opinion is that it is appropriate to use Student's t-test for analyzing the data shown in this manuscript, because the triplicate data are expected to follow normal distribution. The numbers of n do not matter. Further, many papers analyzing results of similar experiments use Student's t-test. We could easily search papers using student's t-test for analyzing cGAS-STING-induced responses in a triplicate manner like following papers, Nat Immunol 22, 485–496 (2021), Nat Immunol 21, 727–735 (2020), Nat Immunol 21, 158–167 (2020).

Thus, we believe that the use student's t-test is appropriate for the statistical analysis in the experimental data shown in this manuscript.

It is irrelevant to look at renal pathology 7 days after pristane treatment (Fig. S1). Pristane-treated B6 mice do not develop severe glomerulonephritis to begin with, and the onset of renal disease is delayed for 4-6 months or more after pristane treatment.

We thank the reviewer for the comment. As the reviewer points out, it is well known that the pristane treatment in WT mice induces glomerular nephritis after 4-6 months. In addition, we have reported that aged *Tank*-deficient mice develop glomerular nephritis (Kawagoe et al. 2009). Thus, we analyzed the renal pathology of pristane-treated TANK-deficient mice to eliminate that possibility that pristane accelerates the renal changes in *Tank*-deficient mice, and the acute renal failure is the cause of the lethality. The lack of significant histological changes in pristane-treated kidney indicate that the cause of the lethality is not the acute renal failure, which supports the contribution of DAH in the lethality of *Tank*-deficient mice.

Are the cells in the last column of graph Fig. 4A co-transfected with five different

plasmids (TANK, IRF7, MyD88, TRAF6, and the IFN β promoter-Luc plasmid)? What were the doses of each plasmid, and what was used as controls for these multiple plasmids?

Yes, the data shown in the last column of the Fig. 4A were following the cotransfection of all the indicated plasmids together with the IFN β promoter-luciferase and Renilla control plasmids as the reviewer indicated. We adjusted the total amounts of transfected plasmids by using a mock empty plasmid. We indicated the amounts of plasmids in the new Fig. 4A, and modified the materials and methods section, and the figure legends.

The numbers of mice in Fig. 5A are adequate, but it is again unclear why none of the wild type mice died over a period of 40 days from DAH. Nevertheless, Sting-deficiency improves survival on a Tank-deficient background. However, Tlr7-deficiency also has an effect, although less than that of Sting. Is reversal of the Tank-deficient phenotype just a function of how much interferon is being made? Or, since IL-10 is protective against DAH, could the severity of DAH be related to how much IL-10 is available?

The lethality of pristane-treated WT mice was lower than the previous reports throughout our experiments. Thus, we feel that the difference in the conditions of animal housing can be the cause of the difference.

Also, we thank the reviewer for the comment on the cause of DAH. Since the IFNR deficiency rescued pristane-induced lethality under *Tank* deficiency (Fig. 2A), we believe that type I IFNs are essential. However, as the reviewer points out, it was reported that pristane-induced mortality by DAH was increased in *Il10*-deficient mice (Zhuang H et al. Arthritis & Rheumatology 2017). Although IL-10 is anti-inflammatory, the expression of IL-10 is known to be upregulated by NF- κ B downstream of TLR and STING signaling pathways (O'Garra and co-authors J Immunol 2005, Barber GN and co-authors Cell Reports 2017). Consistent with our previous report showing the function of TANK in suppressing NF- κ B pathway downstream of TLR (Kawagoe T et al. Nat Immunol 2009), the analysis of microarray data using WT and *Tank*-deficient peritoneal macrophages shown in the Figure R1 below revealed that 1) TLR7 (R848) stimulation upregulated the expression of *Il10* mRNA in addition to *Il6* and *Ifnb1* in macrophages. 2) TANK deficiency slightly increased, but not decreased, the expression of *Il10* like *Il6* and *Ifnb1* compared with WT cells. 3) The expression levels of IL-10 receptors, *Il10ra* and *Il10rb*, was comparable between WT and *Tank*-deficient macrophages. These results suggest that TANK deficiency at least does not decrease the

expression of IL-10 or its receptor in macrophages, and IL-10 may not be the target of TANK for controlling DAH.

Figure R1, The expression of *IL10*, *IL10ra*, *IL10rb*, *IL6* and *Ifnb1* in WT and TANK-deficient (KO) macrophages stimulated with R848.

Reviewer #2 (Comments to the Authors (Required)):

Diffuse alveolar hemorrhage (DAH) is a rare complication occurs in an autoimmune disease systemic lupus erythematosus (SLE). Mechanisms that regulate DAH development is incompletely understood. In the current manuscript, the authors focused on identifying the role and mechanism by which TANK, a negative regulator of the NF- κ B signaling may inhibit DAH development. The authors used the pristane induced SLE mouse model, which developed DAH, to understand the mechanisms that may promote DAH. They found that TANK prevents fatal DAH by negatively regulating type I interferon production via the STING pathway. Overall manuscript is significant and novel in understanding the mechanism of DAH development in SLE.

We thank the reviewer for finding this study novel and interesting.

Comments:

(1) Data presented in Figure 5A indicates that both TLR7 and cGAS-STING pathways are involved in DAH development. Pristane is likely to induce the exposure of both DNA- and RAN-based antigens. Therefore, the authors should emphasize the potential contribution of TLR7 signaling to DAH in the discussion.

We thank the reviewer for the suggestion. As the reviewer points out, TLR7 also contributes to the pristane-induced lethality of *Tank*-deficient mice. According to the reviewer's suggestion we discussed the potential roles of TLR7 in the discussion section (lines 371-377).

(2) Although identifying the role of TANK in regulating innate cell infiltration into peritoneal cavity is a good start, similar flow cytometry analysis should be performed for the lung to determine whether TANK is similarly controlling the innate cell infiltration into the lung, the site of DAH.

We thank the reviewer for the suggestion.

It was reported that monocytes and neutrophils infiltrate into the lung causing DAH in response to pristane treatment (Lee PY, JCI Insight 2019). When we analyzed the numbers of Ly6C^{high} monocytes, neutrophils and pDCs in lungs from WT and *Tank*^{-/-} mice by flow cytometry at 7 days after pristane treatment, the numbers of Ly6C^{high} monocytes and neutrophils in the lung were increased in *Tank*^{-/-} mice compared with WT mice (Figure R2 below). The results indicate that TANK controls the infiltration of monocytes and neutrophils in the lung in pristane-treated mice.

Figure R2. Numbers of Ly6C^{high} monocytes, Ly6G⁺Ly6C^{int} neutrophils and CD11c⁺PDCA1⁺ pDCs in PECs from WT and *Tank*^{-/-} mice at 7 days after pristane

treatment.

(3) *The major finding in this manuscript is that TANK prevents DNA-cGAS aggregate formation. The use of TANK^{-/-}-MyD88^{-/-} cells will help solidify TLR-independent role of TANK in DNA-cGAS aggregate formation.*

The reviewer asks whether the DNA-cGAS aggregate formation is independent of TLR or not. To address this question, we used MyD88-deficient cells. As indicated in the Figure R3 below, MyD88 deficiency did not affect the levels of type I IFN induction as well as the formation of DNA-cGAS aggregation in response to the treatment with dsDNA. The observation is consistent with previous reports showing that the recognition of intracellular dsDNA is mediated through cGAS, but not TLRs. In contrast, TLR7-mediated induction of *Ifnb*, ISGs was abrogated in MyD88-deficient macrophages. Thus, we feel it is apparent that the cGAS and TLR systems are controlled by TANK differently.

Figure R3. The expression of *Ifnb1*, *Isg15* and *Cxcl10* in WT, *MyD88*^{-/-}, *Tank*^{-/-} macrophages stimulated with dsDNA and R848.

(4) *Semi-quantification of data presented in Figure 1D and E will be beneficial for the readers as the images are not very clear to show co-localization. Also, there is a typo in Figure 1D- it should be Merge instead of Marge.*

We thank the reviewer for the suggestion. According to the reviewer's suggestion, we quantified the C3 and IgM merged images shown in Fig. 1D by the evaluation of integrated density using ImageJ software, and the data of 5 different images are shown in new Fig. 1E. As show in the new Fig. 1D and 1E, highly augmented deposition of IgM and C3 was observed in lungs from *Tank*^{-/-} mice in response to pristane treatment.

To quantify the endothelial cell death shown in original Fig. 1E (Fig. 1F in the revised manuscript), we performed flow cytometry to quantify TUNEL positive apoptotic endothelial cells. As shown in new Fig. 1G and 1H, the frequency of apoptotic cells in CD45⁺CD31⁺ lung cells was increased in pristane-treated mice under *Tank* deficiency.

We apologize for the typo in the Figure 1D. We corrected it in the Figure.

(5) The authors have not discussed the potential mechanisms for increased autoantibody production in Ifnar2^{-/-}Tank^{-/-} mice.

We thank the reviewer for this valuable suggestion. As the reviewer points out, the depletion of type I IFN signaling did not ameliorate, rather exacerbated autoantibody production in Tank-deficient mice. Although type I IFNs are critical for autoimmunity in various mouse models and even human SLE, there are autoimmunity models which develop independent of type I IFNs. These include experimental autoimmune encephalitis (EAE), DNase II deficiency and TLR7-mediated lupus nephritis mouse models, indicating that type I IFN-independent pathways contribute to the autoimmunity at certain autoimmune conditions. We discussed this issue in the discussion section (lines 378-387).

(6) The authors have presented representative flow plots for effector and memory CD4 T cells and plasma cells in Figure 2E and F. They should include the bar graphs (as in Figure 2C and D) of all mice analyzed.

We thank the reviewer for this suggestion. However, reviewer #1 suggested that the data shown in Figure 2C and 2D are not relevant to the main claim of this study, and therefore we decided to remove these figures according to the reviewer's suggestion.

Reviewer #3 (Comments to the Authors (Required)):

DAH is a serious complication in some SLE patients. Mechanistic insights into DAH pathology is significant and could lead to new intervention strategy. In the manuscript by Wakabayashi A et al., the authors showed that TANK is critical for preventing pristane-induced fatal DAH in mice by suppressing cGAS-STING dependent type I IFNs production. Overall, the study is novel and interesting. Data supporting the main conclusion is generally strong though some quantification will be helpful. The weakness

is related to how TANK influences cGAS production of cGAMP. The current data supporting TANK regulates cGAS ubiquitination is weak. Below are my comments.

We thank the reviewer for evaluating our study novel and interesting. For the concerns raised by the reviewer, we responded to them as following.

Major points:

- 1. In Figure 2A, 5A, the authors clearly showed that type I IFNs and STING drove DAH in TANK^{-/-} mice. The authors may want to examine if type I IFNs neutralizing antibody or STING antagonist H-151 improve survival in pristine treated TANK^{-/-} mice. Such experiments will not only strengthen their argument but also advance a new therapy for DAH.*

We thank the reviewer for the suggestion. The usage of neutralizing antibody to type I IFNs and a STING antagonist is quite interesting experiments to perform in future. However, this study aims to investigate the mechanisms how TANK contributes to pristane-induced DAH, and the use of mice lacking type I IFN receptor and STING showed highly convincing evidence for the roles of the type I IFN signaling and the cGAS-STING pathway. Although we agree that the suggested experiments are intriguing, we feel they are beyond the scope of this manuscript. Nevertheless, we discussed the further use of type I IFN neutralizing antibody and the STING antagonist in the discussion section (lines 399-402).

- 2. SLE shows gender bias. In the methods and figure legend, please specify the sex of the mice used in the experiments.*

We used almost equal numbers of males and females. According to the reviewer's suggestion, we added the information of the sex used in experiments in the Figure legends. Since almost all male and female *Tank*-deficient mice died following pristane treatment, pristane induced fatal DAH irrespective of the gender under TANK deficiency. Furthermore, we did not find a difference in pristane-induced mortality between male and female mice throughout the study.

- 3. Figure 1D and 1E will be benefited from some quantification. Currently, the data is not convincing.*

According to the reviewer's comment, we quantified the integrated density of C3 and IgM merged images by using ImageJ (new Fig. 1E). As shown in Fig. 1E, the C3 and IgM positive regions were significantly increased in *Tank*^{-/-} mouse lungs compared with WT following treatment with pristane. TUNEL positive cells were quantified by using FACS analysis and showed in new Fig. 1G and 1H.

4. *It is not clear if STING expression in Ly6Chi monocyte or pDCs are responsible for type I IFNs production. Can the authors elaborate it in the Discussion? Future experiment using conditional STING knock out mice will be helpful.*

We thank the reviewer for pointing out this interesting issue. According to the reviewer's suggestion, we discuss the cell type(s) in which the STING-dependent pathway is activated in lines 339-344.

5. *STING affected the recruitment of Ly6Chi monocytes (Fig 3) and seems to be critical for DAH. Can the authors discuss how STING may promote monocytes infiltration?*

We thank the reviewer for asking this question. As the reviewer points out, we found that pristane-induced recruitment of Ly6C^{high} monocytes in *Tank*^{-/-} mice depends on the presence of STING. It was reported that the type I IFN signaling induced production of chemokines such as CCL2, CCL7 and CCL12, which recruits Ly6C^{high} monocytes via the interaction with CCR2 on the cells (Lee PY et al. Am J Pathol 2009). Thus, the STING pathway can contribute to the recruitment of Ly6C^{high} monocytes via the production of chemokines activating CCR2 through the production of type I IFNs. We discuss this issue in the Discussion section (lines 333-338).

6. *In line 240, the authors stated " Tank-/- PECs exhibited elevated phosphorylation of TBK1 as well as IRF3 in response to Herring testis DNA stimulation compared with WT cells (Fig. 6C). There is no quantification and the p-IRF3 and p-TBK1 looked comparable.*

According to the reviewer's comments, we quantified the immunoblot data for p-TBK1 and p-IRF3 by using the ImageJ software and normalized to β -actin. As shown in the new Fig. 6C, we indicated relative intensity below the band. Intensities of phospho-TBK1 and phospho-IRF3 were approximately 2 fold higher in *Tank*^{-/-} cells

compared with WT in response to DNA stimulation. These data confirmed that the phosphorylated TBK1 and IRF3 levels were increased in DNA-stimulated *Tank*^{-/-} cells.

7. *In line 273, the authors stated "Indeed, the cGAS-DNA aggregates are co-stained with ubiquitin both in WT and Tank-/- macrophages (Fig. 7E), implying that TANK restricts cGAS-DNA aggregate formation by suppressing ubiquitination in the complex." I do not understand the logic of this statement. If both WT and TANK-/- are co-stained with ubiquitin, how can you draw the conclusion that TANK suppresses the ubiquitination of cGAS?*

We thank the reviewer for the constructive criticism. As indicated in Fig. 7E and new Fig. 7F, the numbers of cGAS-ubiquitin aggregates were significantly increased in cells lacking *Tank*. Since TANK potentially suppresses ubiquitination of proteins, we speculate that the increase of ubiquitinated cGAS in *Tank*-deficient cells may lead to the increase of cGAS-ubiquitin aggregates. We modified our explanation in the results section.

8. *Fig 7E is not convincing. Without quantification, it seems that TANK-/- macrophage have more overall ubiquitination than the WT upon DNA stimulation. It is questionable that TANK will be the predominant factor suppressing DNA-induced ubiquitination in these macrophage.*

We thank the reviewer for the suggestion. According to the reviewer's comment, we quantified the numbers of cGAS and ubiquitin aggregates in 4 (WT) and 6 (*Tank*^{-/-}) independent experiments (new Fig. 7F). As shown in Fig. 7F, the numbers of cGAS-ubiquitin aggregates were significantly increased in *Tank*^{-/-} macrophages compared with WT cells. These data suggest that TANK contributes to the restriction of cGAS-ubiquitin aggregates in macrophages.

9. *Again, without quantification, in Fig 7E, the AF568-cGAS stain seemed much stronger in Tank-/- cells than WT cells, which will indicate that cGAS expression increased in TANK-/- macrophage compared to WT cells. This contradicted to Fig 7B showing cGAS expression is not altered in TANK-/- cells. In all, the authors need substantial more data, not just a Fig 7E, to support their argument that Tank influences cGAS ubiquitination.*

We thank the reviewer for the criticism. As aforementioned, we quantified the cGAS-ubiquitin aggregates and found that the numbers of cGAS-ubiquitin aggregates were significantly increased in *Tank*^{-/-} macrophages compared with WT cells (new Fig. 7F).

We believe that the expression levels of cGAS were not altered between WT and *Tank*^{-/-} macrophages as shown in Fig. 7B. It was reported that cGAS forms lipid droplets with DNA, and cGAS-DNA liquid phase separation highly increases fluorescent intensity of fluorescently labeled cGAS (Du M et al. Science 2018). Therefore, we think that the AF568-cGAS is highly detectable by the microscope only when they are aggregated in response to DNA stimulation as we found in Fig. 7C.

We agree that the further studies are required to demonstrate that TANK controls cGAS ubiquitination. Therefore, we mention the possibility that TANK suppresses cGAS signaling independent of ubiquitination in the Discussion section (lines 367-368).

10. Figure S2A, the authors' interpretation of the data is not accurate. The precise interpretation for Fig S2A is "TANK does not interact with cGAS in HEK cells." The authors did not examine TANK and cGAS association in Ly6Chi monocytes or pDCs where it may influence cGAMP production.

According to the reviewer's comment, we removed the following sentence in the Results section. "These results suggest that TANK suppresses dsDNA-induced type I IFN responses by restricting aggregation of cGAS-DNA in an indirect manner."

11. In Figure S2B, there was no positive control for cGAS-HA IP. Thus, the authors can not conclude that cGAS did not pull down TANK during DNA stimulation.

We thank the reviewer for this comment. In Figure S2B, we show that cGAS-HA failed to co-precipitate TANK-Flag. As the reviewer points out, we also agree that the positive control for the cGAS-binding protein is absent. However, we confirmed the successful immunoprecipitation by checking the cGAS-HA after IP with anti-HA Ab (Fig. S7B). Reciprocally, we show that TANK-Flag did not co-precipitate cGAS-HA as shown in Figure S2A. In this experiment, we confirmed the co-precipitation of TBK1-HA with TANK-Flag as the positive control (Fig. S2A). These results suggest that the immunoprecipitation experiments were appropriately performed.

To precisely describe the experimental system and the results, we explained the

details of experiments for Figures S2A and S2B in the results section.

Minor points:

1. *Figure 2, Infar2^{-/-} should be Ifnar2^{-/-}*
2. *Figure 2E, the Y-axis should be CD62L.*
3. *Figure 2F, the CD138⁺ plasma cells gate should be B220^{dull}, excluding B220^{hi} population.*

We thank the reviewers for pointing out these typos. We corrected accordingly or withdrawn the figure 2E and 2F based on Reviewer 1' comment.

4. *Figure 4A HEK cells experiment showed that TANK inhibit IRF7, MyD88 and TRAF6 production of IFN β . This is a different mechanism for TANK in DAH where the authors propose that TANK inhibits type I IFNs by suppressing cGAS production of cGAMP. To avoid confusion, better remove Fig 4A.*

We thank the reviewer for this comment. Although the contribution of TLR7 and MyD88 in pristane-induced DAH is rather modest, the absence of TLR7 or MyD88 significantly ameliorated pristane-induced lethality in TANK-deficient mice. Therefore, we think it is also important to show how TANK works in the TLR signaling in activating type I IFN responses.

November 10, 2021

RE: Life Science Alliance Manuscript #LSA-2021-01067-TR

Prof. Osamu Takeuchi
Kyoto University
Department of Medical Chemistry, Graduate School of Medicine
Yoshida-Konoe-cho, Sakyo-ku
Kyoto 6068501
Japan

Dear Dr. Takeuchi,

Thank you for submitting your revised manuscript entitled "TANK prevents IFN-dependent fatal diffuse alveolar hemorrhage by suppressing DNA-cGAS aggregation". We would be happy to publish your paper in Life Science Alliance pending final revisions necessary to meet our formatting guidelines. Please also address Reviewer 3's remaining points.

- please add a Category for your manuscript in our system
- please upload both main and supplementary figures as single files
- please use the [10 author names, et al.] format in your references (i.e. limit the author names to the first 10)
- please add your main and supplementary figure legends to the main manuscript text after the references section
- please add a callout for Figure 7F to your main manuscript text

FIGURE CHECKS:

- scale bars in Figures 1C, D, F, and S1F are not readable. Please indicate the size of the scale bars in the appropriate figure legend.
- please add molecular weights next to all protein blots

A. FINAL FILES:

B. MANUSCRIPT ORGANIZATION AND FORMATTING:

Sincerely,

Reviewer #1 (Comments to the Authors (Required)):

The authors have adequately addressed my concerns. The quantification shown in Figures 1 E and H is a big improvement. However, the original immunofluorescence images (Figures D and F) are still quite weak and may not reproduce well.

Reviewer #2 (Comments to the Authors (Required)):

This is a revised manuscript. The authors have adequately addressed my comments and therefore currently this reviewer has no further concerns or quarries.

Reviewer #3 (Comments to the Authors (Required)):

In the revision, the authors addressed some of my previous comments. However, the following major points remain unsolved. As such, the conclusion that TANK suppressing DNA-cGAS aggregation is premature.

Point 3: Figure 1F and 1G are not convincing. In Figure 1G, what was the justification for the exclusion of the FITC-hi population in WT pristane-treated samples (upper right panel)? Figure 1H needed to be re-calculated based on appropriate TUNEL-FITC gating strategy.

Point 6. The pTBK1 and pIRF3 needed to be normalized to TBK1 and IRF3 respectively, not to Actin. It is unknown if TANK^{-/-} cell have similar TBK1 and IRF3 expression level as the WT. Furthermore, showing the pSer366 of STING activation here will strengthen the argument that STING mediates the inflammation in TANK^{-/-} cells.

Point 9. If cGAS liquid droplet formation influences florescent intensity, the authors need to use a second approach to verify cGAS ubiquitination in TANK^{-/-} cells, e.g western blot for ubiquitination detection. Currently, the conclusion that "TANK inhibits generation of DNA-cGAS aggregates harboring ubiquitination" is premature.

Point 11. The authors did not address my question. Figure S2B examined TANK association with cGAS during DNA stimulation. Figure S2B has no positive control. Why do not the authors add TBK1-HA as a positive control for Figure S2B as they did for

Figure S2A? The authors can not conclude that cGAS did not pull down TANK during DNA stimulation.

We thank the reviewers for their constructive criticisms. Below, please find our point-by-point responses to all comments and questions. We would like to thank all the reviewers for being interested in our manuscript and for their thoughtful suggestions.

Reviewer #1 (Comments to the Authors (Required)):

The authors have adequately addressed my concerns. The quantification shown in Figures 1 E and H is a big improvement. However, the original immunofluorescence images (Figures D and F) are still quite weak and may not reproduce well.

We thank the reviewer for the criticism. However, we believe that our data are convincing. Furthermore, the quantification of the images was performed using ImageJ software with the Cell Counter plugin, which showed clear difference in the activation of complements in pristane-treated *Tank*^{-/-} mice.

Reviewer #2 (Comments to the Authors (Required)):

This is a revised manuscript. The authors have adequately addressed my comments and therefore currently this reviewer has no further concerns or quarries.

We thank the reviewer for accepting our manuscript.

Reviewer #3 (Comments to the Authors (Required)):

Point 3: Figure 1F and 1G are not convincing. In Figure 1G, what was the justification for the exclusion of the FITC-hi population in WT pristane-treated samples (upper right panel)? Figure 1H needed to be re-calculated based on appropriate TUNEL-FITC gating strategy.

We thank the reviewer for the suggestion. According to the reviewer's comment, we enlarged the gates for detecting TUNEL-FITC positive cells. As shown in new Figure 1G and 1H, TUNEL positive cells were more *TANK*^{-/-} mice than controls in following pristane treatment even in the revised gating strategy.

Point 6. The pTBK1 and pIRF3 needed to be normalized to TBK1 and IRF3 respectively, not to Actin. It is unknown if TANK^{-/-} cell have similar TBK1 and IRF3 expression level as the WT. Furthermore, showing the pSer366 of STING activation here will strengthen the argument that STING mediates the inflammation in TANK^{-/-} cells.

We thank the reviewer for the comments. Although the reviewer recommended us to use TBK1 and IRF3 for normalization, it is well known that the expression levels of TBK1 or IRF3 do not alter in response to dsDNA stimulation. Given that the Actin levels are not altered both in WT and *TANK*^{-/-} cells throughout the time course after dsDNA stimulation, we believe that the current data are sufficient to demonstrate that dsDNA-induced phosphorylation of TBK1 and IRF3 were elevated *Tank*^{-/-} macrophages compared with WT cells. We agree that the examination of pSer366 STING is also intriguing, though we feel this is beyond the scope of this manuscript.

Point 9. If cGAS liquid droplet formation influences florescent intensity, the authors need to use a second approach to verify cGAS ubiquitination in TANK^{-/-} cells, e.g western blot for ubiquitination detection. Currently, the conclusion that "TANK inhibits generation of DNA-cGAS aggregates harboring ubiquitination" is premature.

We thank the reviewer for the suggestion. Although we agree that the additional experimental approach will further strength the contribution of TANK in inhibiting cGAS ubiquitination, the current data clearly show that the numbers of cGAS-ubiquitin aggregates were significantly increased in *Tank*^{-/-} macrophages compared with WT cells. Nevertheless, we agree that our claim needs further clarification, and we changed the subtitle of the results section to “TANK may inhibit generation of DNA-cGAS aggregates harboring ubiquitination” (line 249) to weaken the claim.

Point 11. The authors did not address my question. Figure S2B examined TANK association with cGAS during DNA stimulation. Figure S2B has no positive control. Why do not the authors add TBK1-HA as a positive control for Figure S2B as they did for Figure S2A? The authors can not conclude that cGAS did not pull down TANK during DNA stimulation.

We thank the reviewer for the suggestion. We used HEK293 cells for experiments shown both in Figures S2A and S2B. The protocols for transfection, cell lysis, immunoprecipitation and immunoblot analysis are common between them. Thus, we believe that the association between TANK and TBK1 shown in Figure S2A is sufficient to show the positive control for the association between TANK with other protein.

November 12, 2021

RE: Life Science Alliance Manuscript #LSA-2021-01067-TRR

Prof. Osamu Takeuchi
Kyoto University
Department of Medical Chemistry, Graduate School of Medicine
Yoshida-Konoe-cho, Sakyo-ku
Kyoto 6068501
Japan

Dear Dr. Takeuchi,

Thank you for submitting your Research Article entitled "TANK prevents IFN-dependent fatal diffuse alveolar hemorrhage by suppressing DNA-cGAS aggregation". It is a pleasure to let you know that your manuscript is now accepted for publication in Life Science Alliance. Congratulations on this interesting work.

DISTRIBUTION OF MATERIALS:

Again, congratulations on a very nice paper. I hope you found the review process to be constructive and are pleased with how the manuscript was handled editorially. We look forward to future exciting submissions from your lab.

Sincerely,
